# Capturing electron-driven chiral dynamics in UV-excited molecules

Vincent Wanie[1✉], Etienne Bloch[2], Erik P. Månsson[1], Lorenzo Colaizzi[1,3,10], Sergey Ryabchuk[3,4], Krishna Saraswathula[1,3], Andres F. Ordonez[5,11], David Ayuso[5,6,11], Olga Smirnova[6,7], Andrea Trabattoni[1,8], Valérie Blanchet[2], Nadia Ben Amor[9], Marie-Catherine Heitz[9], Yann Mairesse[2], Bernard Pons[2✉] & Francesca Calegari[1,3,4✉]

Chiral molecules, used in applications such as enantioselective photocatalysis[1], circularly polarized light detection[2] and emission[3] and molecular switches[4,5], exist in two geometrical configurations that are non-superimposable mirror images of each other. These so-called (R) and (S) enantiomers exhibit different physical and chemical properties when interacting with other chiral entities. Attosecond technology might enable influence over such interactions, given that it can probe and even direct electron motion within molecules on the intrinsic electronic timescale[6] and thereby control reactivity[7–9]. Electron currents in photoexcited chiral molecules have indeed been predicted to enable enantiosensitive molecular orientation[10], but electron-driven chiral dynamics in neutral molecules have not yet been demonstrated owing to the lack of ultrashort, non-ionizing and perturbative light pulses. Here we use time-resolved photoelectron circular dichroism (TR-PECD)[11–15] with an unprecedented temporal resolution of 2.9 fs to map the coherent electronic motion initiated by ultraviolet (UV) excitation of neutral chiral molecules. We find that electronic beatings between Rydberg states lead to periodic modulations of the chiroptical response on the few-femtosecond timescale, showing a sign inversion in less than 10 fs. Calculations validate this and also confirm that the combination of the photoinduced chiral current with a circularly polarized probe pulse realizes an enantioselective filter of molecular orientations following photoionization. We anticipate that our approach will enable further investigations of ultrafast electron dynamics in chiral systems and reveal a route towards enantiosensitive charge-directed reactivity.

The temporal resolution provided by attosecond technologies developed in the past 23 years gives access to some of the fastest electronic dynamics of matter on their natural timescale. Seminal pump–probe experiments using attosecond light pulses have revealed valence electron dynamics in atoms[16], autoionization dynamics in molecules[17], photoionization delays in solids[18], as well as electron-driven charge migration in simple ionized biomolecules[9,19]. In all of these cases, the intrinsically high photon energy of the attosecond light sources inevitably leads to ionization of the target, which has restricted the measurements to ultrafast dynamics of cationic states.

When aiming to investigate the ultrafast light-induced electron dynamics of chiral molecules in their neutral states, the pump pulse must thus have a photon energy below the ionization threshold and a broadband energy spectrum that can trigger coherent electron motion among several electronic states, and a time duration that ensures prompt excitation before any nuclear motion can take place, together with sufficient temporal resolution. The low ionization potential of most molecular systems thus restricts the choice of the pump pulse wavelength to the UV and vacuum-UV ranges, that is, to a spectral region that cannot trigger intricate high-order, strong-field multiphoton-driven processes[20,21] that rarely occur with natural light sources. When these requirements are met, measurements with high time resolution are possible using pump–probe spectroscopic techniques that are highly sensitive to chirality, such as TR-PECD[12,15] recently used to probe nuclear dynamics, internal conversion and photoionization delays in chiral molecules[11–14,22].

We use ultrashort UV pump pulses[23,24] in combination with circularly polarized near-infrared (NIR) probe pulses to study coherent electronic dynamics in chiral neutral molecules with unprecedented temporal resolution. We apply the chiroptical method of TR-PECD to investigate electron-driven chiral interactions in neutral methyl lactate ($C_4H_8O_3$). Figure 1a,b shows an overview of the experimental approach. First, a linearly polarized UV pulse promptly launches a coherent electronic wave packet just below the ionization threshold in (S)-methyl

[1]Center for Free-Electron Laser Science CFEL, Deutsches Elektronen-Synchrotron DESY, Hamburg, Germany. [2]Université de Bordeaux - CNRS - CEA, CELIA, UMR5107, Talence, France. [3]Physics Department, Universität Hamburg, Hamburg, Germany. [4]The Hamburg Centre for Ultrafast Imaging, Universität Hamburg, Hamburg, Germany. [5]Department of Physics, Imperial College London, London, UK. [6]Max-Born-Institut, Berlin, Germany. [7]Technische Universität Berlin, Berlin, Germany. [8]Institute of Quantum Optics, Leibniz Universität Hannover, Hannover, Germany. [9]CNRS, UPS, LCPQ (Laboratoire de Chimie et Physique Quantiques), FeRMI, Toulouse, France. [10]Present address: Department of Physics, Politecnico di Milano, Milano, Italy. [11]Present address: School of Physical and Chemical Sciences, Queen Mary University of London, London, UK. ✉e-mail: vincent.wanie@desy.de; bernard.pons@u-bordeaux.fr; francesca.calegari@desy.de

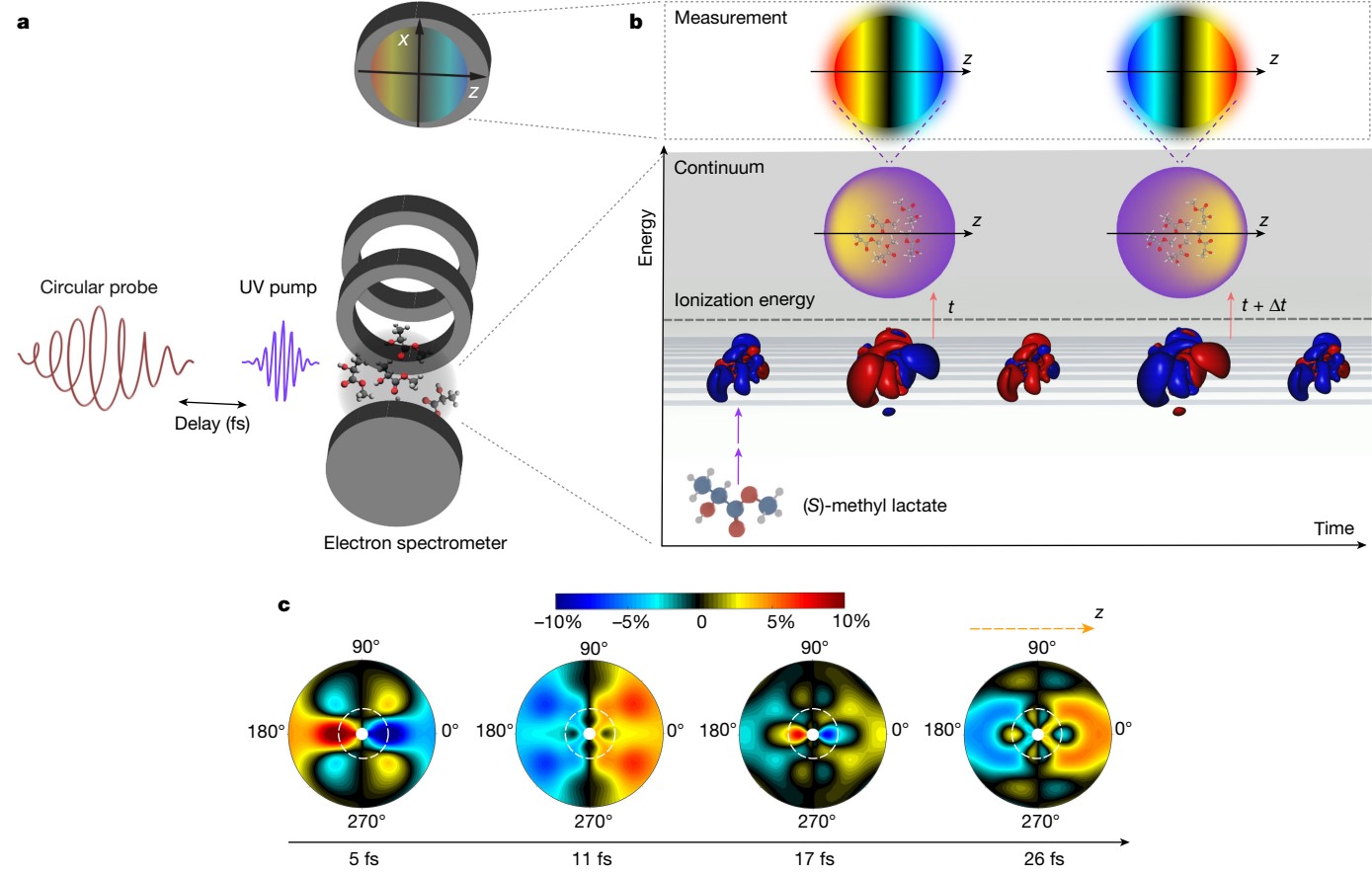

**Fig. 1 | Light-induced chiral dynamics of methyl lactate. a**, A few-femtosecond linearly polarized UV pulse excites an ensemble of randomly oriented chiral molecules, creating an electronic wave packet of Rydberg states by means of two-photon absorption. The dynamics is probed by means of one-photon ionization by a time-delayed circularly polarized NIR pulse. The probing step leads to the ejection of photoelectrons along the light-propagation axis defined along the $z$ direction and the resulting angular distribution is recorded by a VMIS. **b**, The red and blue structures show the temporal evolution of the coherent electron density in the excited neutral molecule: the chiral evolution

of the photoexcited Rydberg wave packet leads to a reversal of the 3D photoelectron angular distribution at two distinct time delays, $t$ and $t + \Delta t$, captured by the measurements. **c**, For each time delay, an image is recorded for both left and right circular polarization of the probe pulse. The differential image PECD$(\varepsilon, \theta, t)$, defined in the main text, is shown for time delays of 5, 11, 17 and 26 fs for photoelectrons with kinetic energies from 25 to 300 meV along the radial coordinate. The white dashed circles identify the photoelectrons below 100 meV that experience an ultrafast reversal of their emission direction in the laboratory frame.

lactate by means of a two-photon transition. Then, a time-delayed circularly polarized NIR probe triggers ionization from the transient wave packet, providing an exceptional instrument response function of $2.90 \pm 0.06$ fs (Extended Data Fig. 1). For each pump–probe delay $t$, the 2D-projected photoelectron angular distributions (PADs) $S^{(h)}(\varepsilon, \theta, t)$ are collected with a velocity map imaging spectrometer (VMIS), for both left ($h = +1$) and right ($h = -1$) circular polarizations of the probe pulse. $\varepsilon$ and $\theta$ stand for the kinetic energy and direction of ejection of the photoelectron in the $(x, z)$ VMIS detection plane, respectively. The chiroptical response is characterized by a photoelectron circular dichroism (PECD) image defined as the normalized difference PECD$(\varepsilon, \theta, t) = 2 \frac{S^{(+1)}(\varepsilon, \theta, t) - S^{(-1)}(\varepsilon, \theta, t)}{S^{(+1)}(\varepsilon, \theta, t) + S^{(-1)}(\varepsilon, \theta, t)}$, subsequently fitted using a pBasex inversion algorithm[11]. Snapshots of the measured PECD$(\varepsilon, \theta, t)$ are presented in Fig. 1c. The signal reaches values of up to 10%, typical for PECD and about two orders of magnitude higher than the analogously defined $g$ factor generally obtained in circular dichroism[25,26]. Low-energy electrons ($\varepsilon \leq 100$ meV; see white dashed circles) are preferentially emitted in the $\theta = 180°$ backward hemisphere at $t = 5$ fs and preferentially ejected forward at $t = 11$ fs. Their main direction of ejection reverses again at $t = 17$ fs. Higher-energy electrons ($\varepsilon > 100$ meV) are more likely emitted forward than backward for $t \geq 11$ fs, but the magnitude of their asymmetry depends on $t$.

The PECD$(\varepsilon, \theta, t)$ images provide quantitative fingerprints of an ultrafast dynamics taking place on the few-femtosecond timescale. To further characterize the temporal evolution of the observed dynamics, we decompose the PAD images in series of Legendre polynomials, $S^{(h)}(\varepsilon, \theta, t) = \sum_{n=0}^{6} b_n^{(h)}(\varepsilon, t) P_n(\cos\theta)$, and calculate the multiphoton PECD (MP-PECD)[27], defined as the normalized difference of electrons emitted in the forward and backward hemispheres for $h = +1$, as MP-PECD$(\varepsilon, t) = 2\beta_1^{(+1)}(\varepsilon, t) - \frac{1}{2}\beta_3^{(+1)}(\varepsilon, t) + \frac{1}{4}\beta_5^{(+1)}(\varepsilon, t)$, in which $\beta_n^{(+1)}(\varepsilon, t) = \frac{b_n^{(+1)}(\varepsilon, t)}{b_0^{(+1)}(\varepsilon, t)}$ (see Methods and Extended Data Figs. 2 and 3). $\beta_1^{(+1)}(\varepsilon, t)$ refers to the isotropic part of the asymmetry in each hemisphere, whereas $\beta_3^{(+1)}(\varepsilon, t)$ encodes anisotropic features owing to pump excitation[11,28], leading to the angular shaping of the PECD illustrated in Fig. 1c. $\beta_5^{(+1)}(\varepsilon, t)$ has been found to be negligible in our measurements. Figure 2a shows MP-PECD$(\varepsilon, t)$ and its $b_1^{(+1)}(\varepsilon, t)$ component is shown in Fig. 2b. The results are shown for ($S$)-methyl lactate and a mirroring symmetric measurement in ($R$)-methyl lactate clearly confirms the chiral character of the Rydberg-induced dynamics, with minor discrepancies owing to slightly lower enantiopurity and statistics (Extended Data Fig. 4). We observe that the unnormalized MP-PECD$(\varepsilon, t)$ behaviour in Fig. 2a closely matches $b_1^{(+1)}(\varepsilon, t)$ in Fig. 2b, indicating that the anisotropic effects included in $\beta_3^{(+1)}(\varepsilon, t)$ play a minor role. The MP-PECD can be partitioned into three kinetic energy ranges, as

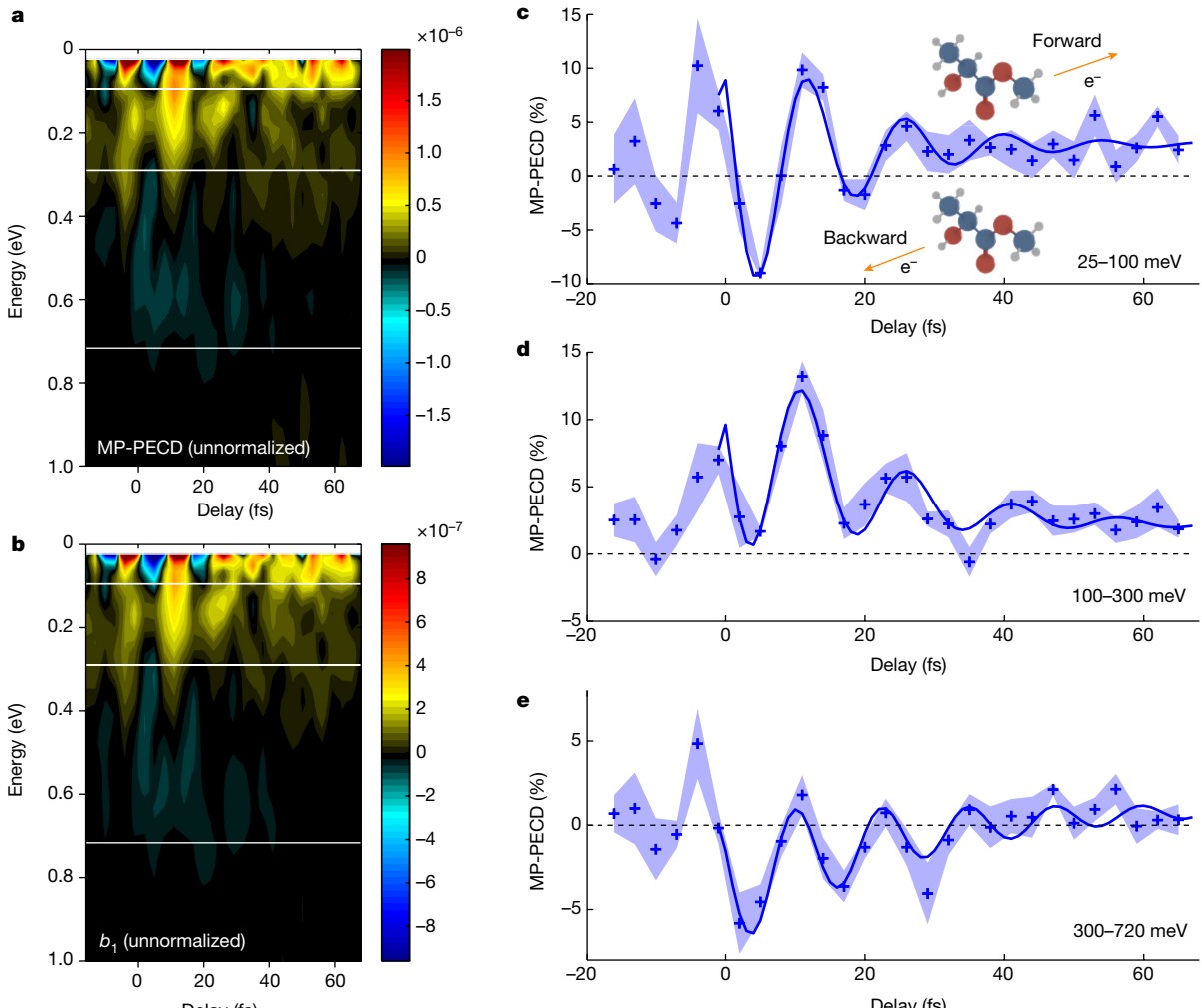

**Fig. 2 | Energy-resolved analysis. a–e**, Temporal evolution of the unnormalized MP-PECD in (S)-methyl lactate (**a**) and corresponding $b_1^{(+1)}$ coefficient (**b**). The white lines identify three different kinetic energy ranges of photoelectrons: 25–100 meV (**c**), 100–300 meV (**d**) and 300–720 meV (**e**). The standard error of the mean over five measurements is shown by the shaded areas. The solid blue lines show the fit of the oscillations from $t = 0$ fs (see the corresponding Fourier analysis in Fig. 3c,e). The change of sign in **c** identifies a reversal of the photoelectron emission direction in the laboratory frame.

identified in Fig. 2a,b. Between 25 and 100 meV, the photoelectron emission asymmetry reverses in about 7 fs (see Fig. 2c). A clear modulation of the asymmetry remains over several tens of femtoseconds, which is also observed at higher $\varepsilon$ between 100 and 300 meV (Fig. 2d) and 300 and 720 meV (Fig. 2e). These modulations are also visible in the time-resolved photoelectron yield $b_0(\varepsilon, t)$, albeit their contrast is considerably weaker (Extended Data Fig. 5). This highlights the capabilities of TR-PECD, which relies on differential measurements, over conventional photoelectron spectroscopy. In the following, we aim at assigning the origin of the fast temporal modulation of the asymmetry, which could involve electronic and/or nuclear degrees of freedom.

We modelled the experiment including both the two-photon UV excitation and the NIR photoionization steps as sequential perturbative processes, within the frozen-nuclei approximation. A detailed description of the theoretical model is provided in Methods. The electronic spectrum of methyl lactate and the two-photon excitation amplitudes are obtained through time-dependent density functional theory[29]. Ionization from the excited states is described using the continuum multiple scattering Xα approach[30,31].

We present the results of our calculations in Fig. 3. The pump pulse populates excited states mainly originating from excitation of the highest occupied molecular orbital (HOMO) of the methyl lactate ground state (see Supplementary Information section 1 and Supplementary

Fig. 2). Figure 3a shows the two-photon excitation cross-section associated with almost pure HOMO excitation to Rydberg states. Subsequent photoionization by the probe pulse leads to the emission of photoelectrons with kinetic energies $\varepsilon = 250$ and 500 meV. These $\varepsilon$ values are representative of the second and third energy ranges discriminated in the experimental data, respectively—the case of low-energy photoelectron dynamics ($\varepsilon = 50$ meV) is discussed in Supplementary Information section 2.3 and illustrated in Supplementary Fig. 5. Including the HOMO excited states of Fig. 3a in the dynamical calculations yields the time-resolved MP-PECD shown in Fig. 3b,d. The calculations are started at $t = 10$ fs to ensure no temporal overlap between the pump and probe pulses. The computed asymmetry presents clear modulations as a function of the pump–probe delay. The power spectra of the MP-PECD signals, obtained by Fourier analysis, are compared with their experimental counterparts in Fig. 3c,e. An excellent agreement is found at $\varepsilon = 250$ meV, at which the oscillatory pattern of the MP-PECD is traced back to the pump-induced coherent superposition of 3d and 4p Rydberg states respectively located at $E_{3d} = 8.834$ eV and $E_{4p} = 9.120$ eV in Fig. 3a. This coherent superposition leads to quantum beatings with approximately 15 fs period, associated with an energy difference between the states of about 300 meV, which survive long after the pump pulse vanishes. We note that the most stable geometries of methyl lactate do not have any vibrational mode in the vicinity of

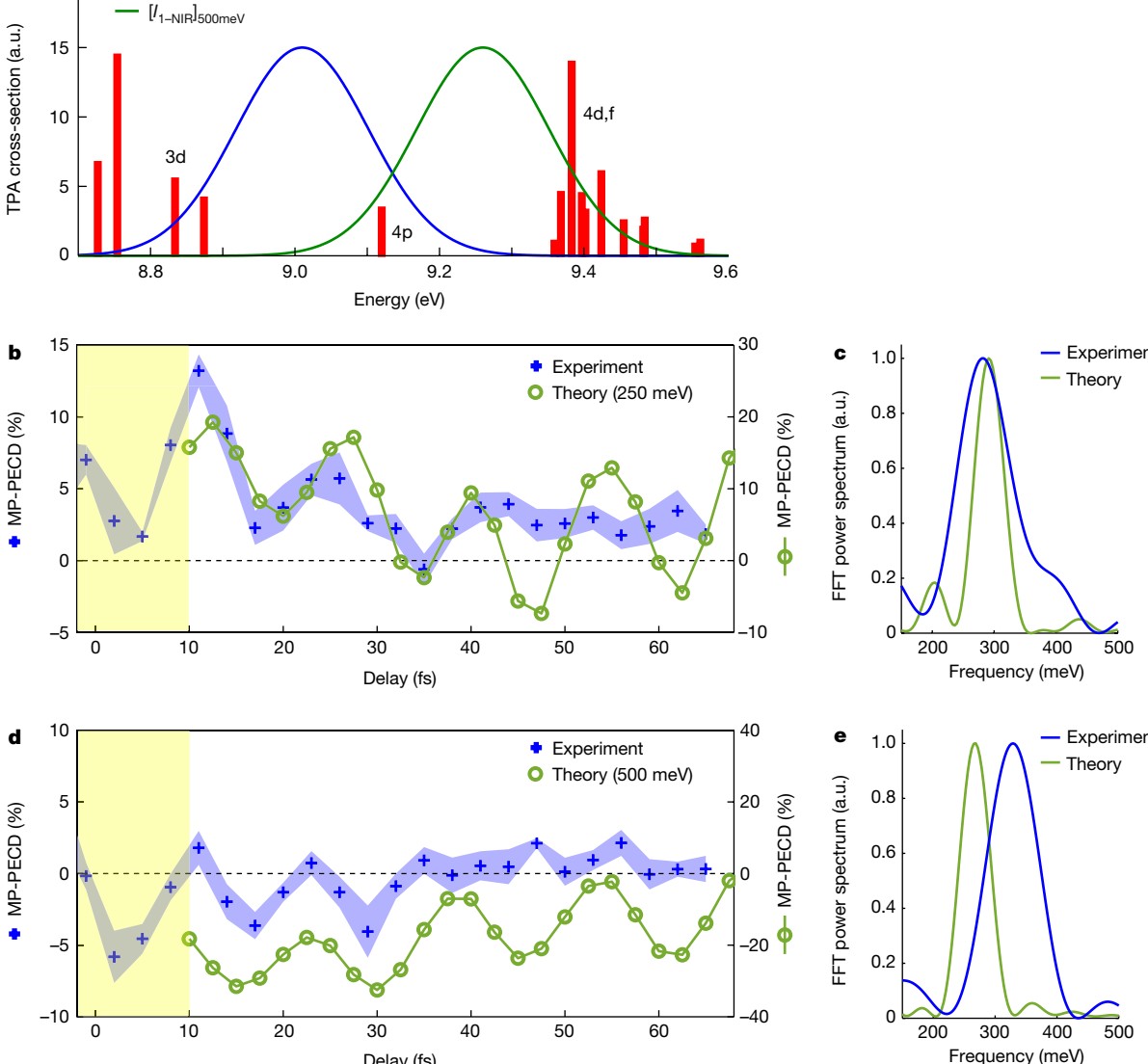

**Fig. 3 | Modelling of the experiment. a**, Two-photon absorption (TPA) cross-sections for the excited states originating from almost pure HOMO excitation. The cross-sections have been convoluted with the UV-pump intensity squared. The blue and green curves correspond to the spectral probe intensity ($I_{1-NIR}$), down-shifted in energy to elicit the transient Rydberg states leading to photoelectrons with energies $\varepsilon = 250$ meV and $\varepsilon = 500$ meV through ionization by one photon centred at frequency $\omega = 1.75$ eV. **b**, Calculated MP-PECD for photoelectrons with $\varepsilon = 250$ meV (green) compared with the experiment (blue). The calculations start at $t = 10$ fs, corresponding to the end of the pump–probe overlap region (yellow area). **c**, Corresponding power spectra from a Fourier analysis. The frequency axis is shown for beatings of excited states with an energy spacing between 150 meV (27.6 fs period) and 500 meV (8.3 fs period). The main peak from the computed MP-PECD evolution is at 291 meV (14.2 fs). The power spectrum of the experimental data, evaluated up to $t = 35$ fs, at which the oscillations are damped, shows a peak frequency at 280 meV (14.8 fs). **d**, Calculated MP-PECD for photoelectrons with $\varepsilon = 500$ meV (green) compared with the experiment (blue). **e**, Corresponding power spectra with a central component at 269 meV (15.4 fs) for the computed curve. The power spectrum of the experimental data is shown, with a central frequency at about 329 meV (12.6 fs). a.u., arbitrary units; FFT, fast Fourier transform.

2,200 cm$^{-1}$ (about 15 fs)[32]. Similarly, the coherent superposition of 4p and 4d,f Rydberg states results in the oscillatory feature of the MP-PECD signal at $\varepsilon = 500$ meV. A small mismatch of about 60 meV is observed between the experimental and theoretical power spectra in Fig. 3e. This mismatch is on the order of the error made in quantum chemistry computations of excited-state energies. Overall, Fig. 3b,d unmistakably demonstrates that the electronic coherence of the intermediate Rydberg states, as identified in Fig. 3a, modulates the molecular chiroptical response.

In our fixed-nuclei description, the electronic coherences leading to oscillatory MP-PECD do not vanish and even lead to an overestimation of the MP-PECD amplitude at all delays $t$. By contrast, the oscillations observed in the experimental MP-PECD (Fig. 2c–e) are damped over

time, which coincides with the approximately 40 fs lifetime encoded in the time-dependent photoelectron yield (Extended Data Fig. 5). Describing the coupled electron and nuclear dynamics in an energy range in which tens of electronic states lie is beyond state-of-the-art theoretical approaches. Therefore, we alternatively performed ab initio molecular dynamics calculations on the ground state of cationic methyl lactate to which all the HOMO Rydberg states involved in the pump–probe dynamics correlate following ionization (see Supplementary Information section 3 and associated Supplementary Figs. 6–8). These calculations suggest that the most probable source of decoherence in this investigation is non-adiabatic transitions.

The oscillations of the chiroptical response for $\varepsilon = 250$ meV mainly result from the coherent superposition of two states, the 3d and 4p

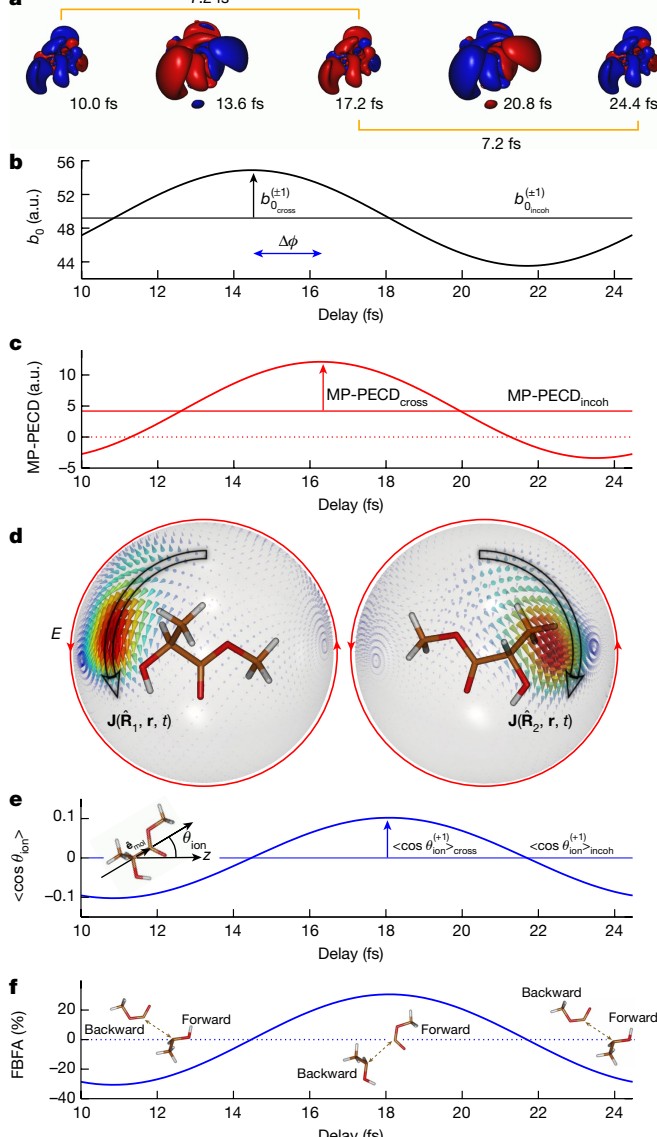

**Fig. 4 | Electron-driven dynamics for a quantum beating of (3d, 4p) Rydberg states monitored at $\varepsilon = 250$ meV. a**, Temporal evolution of the coherent part of the electron density over one period of the quantum beating (see equation (1)). **b**, Photoelectron yield as a function of the pump–probe delay, oscillating in phase with the variation of the electron density shown in **a**, as expected from equations (1) and (2). **c**, MP-PECD as a function of the pump–probe delay according to equation (3). **d**, Snapshots of the electronic current induced by the pump pulse, on a Rydberg sphere of 10 a.u. radius surrounding the molecule for two distinct orientations $\hat{\mathbf{R}}_i$. Propensity rules enhance ionization for orientation $\hat{\mathbf{R}}_1$, for which the current co-rotates with the circularly polarized probe field (red arrow). **e**, Active orientation of the produced cations along the light-propagation axis $\hat{\mathbf{z}}$ as a function of time according to equation (6). **f**, Resulting FBFA along $\hat{\mathbf{z}}$ in the reactive fragmentation of methyl lactate cations (see equation (7)). The insets illustrate the preferential directions of emission of $CO_2CH_3$ and $CH_3CHOH^+$ fragments. a.u., arbitrary units.

Rydberg states. We now investigate in more detail the role of these states in the chiroptical response. For a single molecular orientation $\hat{\mathbf{R}}$, the excited electron wave packet reads, at time $t$ after the pump pulse vanishes, $\Phi(\hat{\mathbf{R}}, \mathbf{r}, t) = \sum_{j=3d,4p} A_j(\hat{\mathbf{R}})\Psi_j(\mathbf{r})\exp(-iE_j t/\hbar)$, in which $A_j(\hat{\mathbf{R}})$ are the real two-photon transition amplitudes associated with the excited states $\Psi_j(\mathbf{r})$. The associated electron density can be partitioned as

$$\rho(\hat{\mathbf{R}}, \mathbf{r}, t) = \rho_{\text{incoh}}(\hat{\mathbf{R}}, \mathbf{r}) + \rho_{\text{cross}}(\hat{\mathbf{R}}, \mathbf{r})\cos[(E_{4p} - E_{3d})t/\hbar] \quad (1)$$

in which $\rho_{\text{incoh}}(\hat{\mathbf{R}}, \mathbf{r}) = A_{3d}^2(\hat{\mathbf{R}})\Psi_{3d}^2(\mathbf{r}) + A_{4p}^2(\hat{\mathbf{R}})\Psi_{4p}^2(\mathbf{r})$ and $\rho_{\text{cross}}(\hat{\mathbf{R}}, \mathbf{r}) = 2A_{3d}(\hat{\mathbf{R}})A_{4p}(\hat{\mathbf{R}})\Psi_{3d}(\mathbf{r})\Psi_{4p}(\mathbf{r})$. Figure 4a shows, for one selected orientation $\hat{\mathbf{R}}$, the coherent part $\rho(\hat{\mathbf{R}}, \mathbf{r}, t) - \rho_{\text{incoh}}(\hat{\mathbf{R}}, \mathbf{r})$ of the electron density, oscillating back and forth along the molecular structure with a period $T = 2\pi\hbar/(E_{4p} - E_{3d})$ of 14.4 fs. Ionization of the 3d and the 4p state superposition leads, after averaging over the orientations $\hat{\mathbf{R}}$, to the total photoelectron yield, which can be decomposed similarly to equation (1):

$$b_0^{(\pm1)}(\varepsilon, t) = b_{0_{\text{incoh}}}^{(\pm1)}(\varepsilon) + b_{0_{\text{cross}}}^{(\pm1)}(\varepsilon)\cos[(E_{4p} - E_{3d})t/\hbar]. \quad (2)$$

The computed yield is presented in Fig. 4b for $\varepsilon = 250$ meV, showing how the coherent state superposition leading to $b_{0_{\text{cross}}}^{(\pm1)}(\varepsilon)\cos[(E_{4p} - E_{3d})t/\hbar]$ modulates the incoherent sum $b_{0_{\text{incoh}}}^{(\pm1)}(\varepsilon)$ of individual ionization cross-sections. The unnormalized MP-PECD can in turn be written as:

$$\text{MP-PECD}(\varepsilon, t) = \text{MP-PECD}_{\text{incoh}}(\varepsilon)$$
$$+ \text{MP-PECD}_{\text{cross}}(\varepsilon)\cos\left[\frac{(E_{4p} - E_{3d})t}{\hbar} - \Delta\phi\right] \quad (3)$$

in which the extra phase $\Delta\phi$ arises from the interference of the state-selective continuum partial wave amplitudes building the asymmetry of the photoelectron yield (see Supplementary Information). As usual, this interference is washed out at the level of the total photoelectron yield[30,33]. The temporal evolution of the unnormalized two-state MP-PECD is shown in Fig. 4c, from which we extract the time delay $\Delta t = 1.8$ fs associated with $\Delta\phi = 0.79$ rad. The MP-PECD reverses sign within one period of the oscillation because the asymmetries of single 3d-mediated and 4p-mediated pathways, contributing to the incoherent MP-PECD around which the coherent part oscillates, verify $|\text{MP-PECD}_{\text{cross}}(\varepsilon)| > |\text{MP-PECD}_{\text{incoh}}(\varepsilon)|$. A similar behaviour is observed at lower kinetic energy in the measurement reported in Fig. 2c. Notably, the MP-PECD depends not only on the transient bound resonances—as evidenced in Figs. 2c–e and 3a,b,d—but also on the dichroism encoded by ionization with circularly polarized light. In this respect, we note that a photoexcitation electron circular dichroism (PXECD) configuration[34], in which molecules are photoexcited by a circularly polarized pump pulse and subsequently ionized with a linearly polarized probe, would reduce the degrees of freedom to only the transient bound resonances.

The electron dynamics uncovered in this work underlies yet another consequence on the molecular response of photoexcited chiral systems: the coherent superposition of excited states induced by the pump allows to selectively filter—within a few femtoseconds—specific molecular orientations through enantiosensitive photoionization[10]. An electron moving within the coherent state superposition creates an electronic current[35] $\mathbf{J}(\hat{\mathbf{R}}, \mathbf{r}, t) = \frac{\hbar}{m}\Im[\Phi^*(\hat{\mathbf{R}}, \mathbf{r}, t)\nabla\Phi(\hat{\mathbf{R}}, \mathbf{r}, t)]$, which reduces to

$$\mathbf{J}(\hat{\mathbf{R}}, \mathbf{r}, t) = \frac{\hbar}{m}A_{3d}(\hat{\mathbf{R}})A_{4p}(\hat{\mathbf{R}})[\Psi_{4p}(\mathbf{r})\nabla\Psi_{3d}(\mathbf{r}) - \Psi_{3d}(\mathbf{r})\nabla\Psi_{4p}(\mathbf{r})]$$
$$\sin\left[\frac{(E_{4p} - E_{3d})t}{\hbar}\right] \quad (4)$$

when expanding $\Phi$ on the (real) 3d and 4p bound eigenstates. The chirality of the molecule induces a curl in the generated electron current, whose rotation direction reverses periodically. Rotating electron currents are known to influence the ionization probability by circularly polarized light: the propensity rules[36] establish that one-photon ionization is enhanced when the electrons rotate in the same direction as the electric field. Therefore, the molecules oriented such that their electronic current rotates in the same plane and direction as the ionizing laser pulse are preferentially ionized; see Fig. 4d. Consequently,

the produced molecular cations are selectively oriented along the probe polarization rotation axis, corresponding to the light-propagation axis $\hat{z}$.

To quantify the degree of orientation of the photoionized molecules, we select a unitary vector $\hat{e}_{mol}$ fixed to the internal C–C bond of the methyl lactate cation, as illustrated in the inset of Fig. 4e, and calculate its averaged value over the probe-filtered molecular orientations in the laboratory frame as[10]

$$\langle \hat{e}_{lab} \rangle_{\hat{R}}^{(\pm 1)}(\varepsilon, t) = \frac{\int d\hat{R} W^{(\pm 1)}(\hat{R}, \varepsilon, t) \hat{e}_{lab}(\hat{R})}{b_{0_{incoh}}^{(\pm 1)}(\varepsilon)} \qquad (5)$$

in which $\hat{e}_{lab}(\hat{R})$ is the $\hat{e}_{mol}$ vector passively rotated in the laboratory frame and $W^{(\pm 1)}(\hat{R}, \varepsilon, t)$ is the ionization rate associated to helicity $h = \pm 1$ and photoelectrons of energy $\varepsilon$. Notably, the averaged orientation of the cations depends on $\varepsilon$ because the $\varepsilon$ dependence of the underlying photoionization yield is not the same for all orientations $\hat{R}$. The $x$ and $y$ components of $\langle \hat{e}_{lab} \rangle_{\hat{R}}^{(\pm 1)}(\varepsilon, t)$ are found to be zero and only the $z$ component survives the averaging[10] (see Supplementary Fig. 9), leading to

$$\langle \cos\theta_{ion} \rangle_{\hat{R}}^{(\pm 1)}(\varepsilon, t) = \langle \cos\theta_{ion} \rangle_{cross}^{(\pm 1)}(\varepsilon) \sin[(E_{4p} - E_{3d})t/\hbar] \qquad (6)$$

in which $\theta_{ion}$ is the angle between the internal C–C bond and the probe propagation $\hat{z}$ axis (see inset of Fig. 4e). $\langle \cos\theta_{ion} \rangle_{cross}^{(\pm 1)}(\varepsilon)$ involves chiral-sensitive products of 3d and 4p excitation and ionization amplitudes. The temporal evolution of $\langle \cos\theta_{ion} \rangle_{\hat{R}}^{(+1)}$ is illustrated in Fig. 4e for $\varepsilon = 250$ meV. When $\langle \cos\theta_{ion} \rangle_{\hat{R}}^{(+1)}(\varepsilon, t) < 0$, the $CO_2CH_3$ moiety of the methyl lactate cations preferentially points forward with respect to $\hat{z}$, whereas instead it points backward when $\langle \cos\theta_{ion} \rangle_{\hat{R}}^{(+1)}(\varepsilon, t) > 0$. Such asymmetry could be detected by resolving the direction of fragmentation of the molecular cations (see Supplementary Information section 4.2 and associated Supplementary Figs. 10 and 11). Indeed, the relative numbers of molecules pointing forward and backward at time $t$, $N_+^{(\pm 1)}(\varepsilon, t)$ and $N_-^{(\pm 1)}(\varepsilon, t)$, respectively, can be linked to $\langle \cos\theta_{ion} \rangle_{\hat{R}}^{(\pm 1)}(\varepsilon, t)$ (see Methods). This ultrafast filtering of molecular orientation affects the subsequent reactive dynamics of methyl lactate cations, with prompt photoionization dictating the subsequent dissociation along the selected molecular orientation. A forward/backward fragment asymmetry (FBFA) thus naturally arises, which we define as

$$FBFA^{(\pm 1)}(\varepsilon, t) = 2 \frac{N_+^{(\pm 1)}(\varepsilon, t) - N_-^{(\pm 1)}(\varepsilon, t)}{N_+^{(\pm 1)}(\varepsilon, t) + N_-^{(\pm 1)}(\varepsilon, t)}. \qquad (7)$$

The FBFA is shown in Fig. 4f for $h = +1$ and $\varepsilon = 250$ meV, reaching absolute values of about 30%, whereas its temporal evolution is dictated by the behaviour of the underlying electron current $J(\hat{R}, r, t)$. Similarly to the MP-PECD, the FBFA switches sign for $h = -1$ or when the other enantiomeric form of methyl lactate molecules is considered. Because the FBFA is created by the electron current, it vanishes in the case of incoherent population of excited states (see Supplementary Information section 4). This shows that the chiral electronic coherence directly observed in our experiment through TR-PECD is crucial to achieve control over enantioselective dynamics of the nuclei.

We have taken an important step forward by resolving the coherent chiral electronic dynamics of a chiral molecule in the first instants following prompt excitation by an achiral few-femtosecond UV pulse. The results showcase that TR-PECD can provide insights on the role of the primary electron dynamics in the light-induced chiral response of complex molecules. Beyond its impact on the chiroptical properties of the system, the chiral currents generated in our experiment can be exploited for photochemical control, as exemplified by our calculations on enantiosensitive charge-directed reactivity leading to oriented fragmentation. From a broader perspective, our results contribute to the fundamental understanding of electronic chirality at the molecular level and its impact on primary enantiosensitive interactions.

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

## Methods

### Experimental setup

The experiments were carried out with a 1-kHz titanium:sapphire laser (Femtopower, Spectra-Physics), delivering 25-fs, 12-mJ pulses at 800 nm. 5.6 mJ was used for spectral broadening in a 2.3-m-long hollow-core fibre (Few-Cycle Inc.) filled with a pressure gradient of helium gas. The fibre setup seeds an all-vacuum Mach–Zehnder-like interferometer with 5-fs NIR pulses. One arm is used for the generation of the UV-pump pulse by means of third-harmonic generation in a laser-machined glass cell filled with 7.2 bar of neon gas. A pair of silicon superpolished substrates (Gooch & Housego) is used at Brewster's angle to attenuate the residual part of the NIR driving field by three orders of magnitude while reflecting approximately 16% of the UV radiation (50 nJ). In the second arm of the interferometer, the remaining part of the NIR beam is focused to the experimental region by a toroidal mirror ($f = -900$ mm), followed by a motorized zero-order quarter-wave plate (B. Halle) to control the helicity of the circularly polarized probe pulses (16 μJ), with an intensity of $5 \times 10^{12}$ W cm$^{-2}$. The instrument response function of $2.90 \pm 0.06$ fs is obtained by a global fit of the non-resonant (Gaussian) dynamics of selected ion masses acquired simultaneously with the photoelectron spectra (Extended Data Fig. 1). Liquid (S)-methyl lactate (97% enantiomeric excess; Sigma-Aldrich) was evaporated and transported by diffusion to a VMIS to measure the photoelectron angular distribution as a function of the pump–probe time delay. To avoid condensation of the sample along the transport line within the molecular source, a temperature gradient from 85 °C to 95 °C was applied.

### Analysis of the VMIS images

For each pump–probe delay $t$, the photoelectron angular distributions are collected with a VMIS for both left ($h = +1$) and right ($h = -1$) circular polarizations of the NIR probe pulse to yield $S^{(h)}(\varepsilon, \theta, t)$, in which $\varepsilon$ is the kinetic energy of the photoelectron and $\theta$ its emission angle with respect to the light-propagation axis. The differential PECD image is then defined as the normalized difference $\text{PECD}(\varepsilon, \theta, t) = 2\frac{S^{(+1)}(\varepsilon, \theta, t) - S^{(-1)}(\varepsilon, \theta, t)}{S^{(+1)}(\varepsilon, \theta, t) + S^{(-1)}(\varepsilon, \theta, t)}$, subsequently fitted using a pBasex inversion algorithm[11]. Its evolution is monitored as a function of the pump–probe delay $t$ in Fig. 1c. $S^{(h)}(\varepsilon, \theta, t) = \sum_{n=0}^{2N} b_n^{(h)}(\varepsilon, t) P_n(\cos\theta)$, in which $P_n(\cos\theta)$ are Legendre polynomials and $N = 3$ is the total number of photons absorbed to reach the continuum from the ground state: the pump-induced excitation involves two photons, whereas ionization consists of the absorption of one NIR probe photon. $b_0^{(h)}(\varepsilon, t)$ corresponds to the total (angle-integrated) photoionization cross-section. In the case of a sample of randomly oriented achiral molecules, the PAD is symmetric with respect to the light-propagation axis, so that the $S^{(h)}(\varepsilon, \theta, t)$ expansion is restricted to even $n$. For randomly oriented chiral molecules, the asymmetric contribution to the photoelectron yield emerges from the extra $b_n^{(h)}$ amplitude coefficients with odd $n$. Besides $\text{PECD}(\varepsilon, \theta, t)$, it is convenient to introduce an angularly integrated quantity to characterize the whole chiroptical response at fixed kinetic energy. Defining it as the difference of electrons emitted in the forward and backward hemispheres for $h = +1$, normalized to the average number of electrons collected in one hemisphere, we obtain the so-called MP-PECD[27], $\text{MP-PECD}(\varepsilon, t) = 2\beta_1^{(+1)}(\varepsilon, t) - \frac{1}{2}\beta_3^{(+1)}(\varepsilon, t) + \frac{1}{4}\beta_5^{(+1)}(\varepsilon, t)$, in which $\beta_n^{(+1)}(\varepsilon, t) = \frac{b_n^{(+1)}(\varepsilon, t)}{b_0^{(+1)}(\varepsilon, t)}$. The time-resolved and energy-resolved amplitude coefficients $b_n^{(+1)}(\varepsilon, t)$, together with the resulting unnormalized $\text{MP-PECD}(\varepsilon, t)$, are shown in Extended Data Fig. 2. Note that $\beta_5^{(+1)}(\varepsilon, t)$ is not included because of its negligible contribution to the total signal. A similar analysis protocol, ignoring the lack of cylindrical symmetry induced by the anisotropy of excitation of the linearly polarized UV-pump pulse, was used in refs. 11 and 12. It was recently shown that the harmonic terms describing symmetry breaking could be of significant amplitude[37]. In our experiment, they could be measured by repeating the measurements with a few different orientations of the

pump polarization with respect to the detector plane, and tomographically reconstructing the 3D-PECD by Hankel transform as was done in ref. 37.

### Enantiomeric comparison

To evaluate the robustness of the results, the experiment was repeated in (R)-methyl lactate (96% enantiomeric excess; Sigma-Aldrich). The enantiomer comparison is shown in Extended Data Fig. 4, in which all the main features observed in (S)-methyl lactate are present: the sign of the MP-PECD signal reverses for low-kinetic-energy electrons in Extended Data Fig. 4a and the oscillations in Extended Data Fig. 4b,c are of opposite sign. This mirroring effect validates the quality of the experimental data. The larger error bars for the data in Extended Data Fig. 4a,b are because of the reduced enantiopurity (96%) of (R)-methyl lactate and a quarter of the statistics compared with (S)-methyl lactate. Although this prevents a reliable Fourier analysis for each kinetic energy range of the (R) enantiomer, we show in Extended Data Fig. 4d the excellent agreement in beating frequency at about 329 meV for the MP-PECD of the high-kinetic-energy electrons shown in Extended Data Fig. 4c.

### Analysis of the $b_0(\varepsilon, t)$ coefficient

The average $b_0(\varepsilon, t)$ shown in Extended Data Fig. 5a–c for the three $\varepsilon$ regions of interest defined in Fig. 2a was fitted using an exponentially modified Gaussian distribution[38]:

$$f(t; t_0, A_1, A_2, \sigma, \tau)$$
$$= A_1 \times \text{erfc}\left(\frac{\sigma}{\tau\sqrt{2}} - \frac{(t - t_0)}{\sigma\sqrt{2}}\right) \times \exp\left(-\frac{(t - t_0)}{\tau} + \frac{\sigma^2}{2\tau^2}\right) + A_2$$

in which $t_0$ is the pump–probe overlap time, $\sigma$ the standard deviation of the Gaussian, $\tau$ the exponential lifetime and $A_1$ and $A_2$ are, respectively, amplitude and background constants. A characteristic decay time on the order of 40 fs is obtained, assigned to the population relaxation from the initially populated Rydberg states to lower-lying states.

To extract oscillating features, the fit curve was subtracted from the experimental data and a vertical shift was applied to the resulting curve by subtracting its average value. The corresponding residuals are shown in Extended Data Fig. 5d–f. The Fourier analysis of these experimental residuals is compared with the theoretical results in Extended Data Fig. 5g–i. Although a good agreement can be obtained in Extended Data Fig. 5h (259 versus 279 meV beating frequency for the kinetic energy range $\varepsilon = 100$–300 meV), the strong background signal generally makes the extraction of the residuals less reliable than for the corresponding background-free MP-PECD curves for which a similar analysis has been performed (see Fig. 2 and Extended Data Fig. 4d).

### Computation of TR-PECD

At time $t$ after the pump pulse vanishes, the electron wave packet formed in a methyl lactate molecule whose orientation in the laboratory frame is characterized by $\hat{\mathbf{R}}$ reads

$$\Phi(\hat{\mathbf{R}}, \mathbf{r}, t) = \sum_i \mathcal{A}_i(\hat{\mathbf{R}}) \Psi_i(\mathbf{r}) \exp(-iE_i t/\hbar),$$

in which $\Psi_i(\mathbf{r})$ are excited states with energies $E_i$ and two-photon absorption amplitudes from the ground state $\mathcal{A}_i(\hat{\mathbf{R}})$. These states, energies and transition amplitudes have been obtained by time-dependent density functional theory[29] calculations using the LCBLYP[39] functional to describe electron exchange and correlation. The calculations are detailed in Supplementary Information section 1 and illustrated in Supplementary Figs. 1–3. In the spectral region spanned by the pump pulse, most of the excited states have a Rydberg character and originate from the excitation of the methyl lactate HOMO (see Supplementary Fig. 2).

The absorption of one NIR photon of the probe pulse leads to the ejection of a photoelectron with wavevector $\hat{\mathbf{k}}'$ in the molecular frame. The associated ionization dipole is

$$\mathbf{d}_{\mathbf{k}'}^{h,\mathrm{mol}}(\hat{\mathbf{R}}, t) = \sum_i \mathcal{A}_i(\hat{\mathbf{R}}) \sqrt{I_{1-\mathrm{NIR}}(\omega_i)} < \Psi_{\mathbf{k}'}^{(-)} \, |\hat{\mathbf{e}}_\mathbf{h} \cdot \mathbf{r}| \, \Psi_i > \exp(-iE_i t/\hbar)$$

in which $I_{1-\mathrm{NIR}}(\omega_i)$ is the spectral intensity of the probe pulse at frequency $\omega_i = k'^2/2 + I_\mathrm{p} - E_i$, with $I_\mathrm{p}$ the methyl lactate ionization potential, $\hat{\mathbf{e}}_\mathbf{h}$ the circular polarization of the probe pulse ($h = \pm 1$) and $\Psi_{\mathbf{k}'}^{(-)}$ the ingoing scattering state associated with the electron ejected in the continuum. Neither the scattering state nor the excited states explicitly depend on $t$ because the calculations are made assuming that the nuclei remain frozen at their equilibrium locations at all $t$ (see Supplementary Information section 3). The computation of $\Psi_{\mathbf{k}'}^{(-)}(\mathbf{r})$ requires an explicit form of the exchange potential that the LCBYP functional does not provide. Therefore, we alternatively use the X$\alpha$ approximation for the electron exchange interaction[30,31] to calculate $\Psi_{\mathbf{k}'}^{(-)}(\mathbf{r})$, as detailed in Supplementary Information section 2.2.

Rotating the ionization dipole into the laboratory frame allows us to define the orientation-averaged differential ionization cross-section as

$$\frac{\mathrm{d}\overline{\sigma}^{(h)}}{\mathrm{d}\Omega_\mathbf{k}}(k, \theta, \varphi, t) \propto \int \mathrm{d}\hat{\mathbf{R}} \, |\mathbf{d}_\mathbf{k}^{h,\mathrm{lab}}(\hat{\mathbf{R}}, t)|^2$$

in which $k = k'$ and $(\theta, \varphi)$ are the spherical angles characterizing the direction $\hat{\mathbf{k}}$ of electron ejection in the laboratory frame; $\theta$ is defined with respect to the pulse propagation direction $\hat{\mathbf{z}}$. Although the cross-section can be put in the closed form

$$\frac{\mathrm{d}\overline{\sigma}^{(h)}}{\mathrm{d}\Omega_\mathbf{k}}(k, \theta, \varphi, t) = \sum_{s=0}^{6} \sum_{i=-2}^{2} b_{s,2i}^{(h)}(k, t) Y_s^{2i}(\theta, \varphi),$$

in which $Y_s^{2i}(\theta, \varphi)$ are spherical harmonics, we show in Supplementary Information section 2.2 that the MP-PECD is

$$\mathrm{MP\text{-}PECD}(\varepsilon, t)$$
$$= \frac{1}{b_{0,0}^{(+1)}(\varepsilon, t)} \left( 2\sqrt{3}\, b_{1,0}^{(+1)}(\varepsilon, t) - \frac{\sqrt{7}}{2} b_{3,0}^{(+1)}(\varepsilon, t) + \frac{\sqrt{11}}{4} b_{5,0}^{(+1)}(\varepsilon, t) \right).$$

in which $\varepsilon = \frac{\hbar^2 k^2}{2m}$, with $m$ the electron mass, is the photoelectron kinetic energy. The $b_{s,2i}^{(h)}$ coefficients basically depend on partial-wave ionization amplitudes weighted by the primary excitation factors, as shown in Supplementary Information section 2.2. The convergence of the computed MP-PECD with respect to the number of excited states included in the expansion of the electron wave packet is discussed in Supplementary Information section 2.3 and illustrated in Supplementary Fig. 4 for $\varepsilon = 0.5$ eV.

### Probe-induced active orientation of the sample and enantioselective reactive dynamics

The ionization rate $W^{(\pm 1)}(\hat{\mathbf{R}}, \varepsilon, t)$ involved in the averaged value of the probe-filtered molecular orientation in the laboratory frame, $\langle \hat{\mathbf{e}}_{\mathrm{lab}} \rangle_{\hat{\mathbf{R}}}^{(\pm 1)}(\varepsilon, t)$ in equation (5), is

$$W^{(\pm 1)}(\hat{\mathbf{R}}, \varepsilon, t) \propto \int |\mathbf{d}_\mathbf{k}^{\pm 1,\mathrm{lab}}(\hat{\mathbf{R}}, t)|^2 \, \mathrm{d}\hat{\mathbf{k}}.$$

Its expression, involving primary excitation and ionization amplitudes, is detailed in Supplementary Information section 4.

The $z$ component of $\langle \hat{\mathbf{e}}_{\mathrm{lab}} \rangle_{\hat{\mathbf{R}}}^{(\pm 1)}(\varepsilon, t)$, $\langle \cos\theta_{\mathrm{ion}} \rangle_{\hat{\mathbf{R}}}^{(\pm 1)}(\varepsilon, t)$, is given in equation (6). Its relation with the number of molecules pointing forward and backward with respect to the light-propagation axis, $N_+^{(\pm 1)}(\varepsilon, t)$ and $N_-^{(\pm 1)}(\varepsilon, t)$, respectively, is derived from the simple orientation model introduced in ref. 10: the distribution of oriented molecules is described by the wavefunction $\chi(\theta_{\mathrm{ion}}, \varepsilon, t) = a_0(\varepsilon, t) Y_0^0(\theta_{\mathrm{ion}}) + a_1(\varepsilon, t) Y_1^0(\theta_{\mathrm{ion}})$, in which $Y_0^0(\theta_{\mathrm{ion}})$ and $Y_1^0(\theta_{\mathrm{ion}})$ are the usual spherical harmonics and $a_0^2(\varepsilon, t) + a_1^2(\varepsilon, t) = 1$. Note that higher $(l, m)$ orders are not necessary to describe basically a forward/backward asymmetry. $\langle \cos\theta_{\mathrm{ion}} \rangle_{\hat{\mathbf{R}}}^{(\pm 1)}(\varepsilon, t)$ can be evaluated using the $\chi(\theta_{\mathrm{ion}}, \varepsilon, t)$ expansion as

$$\langle \cos\theta_{\mathrm{ion}} \rangle_{\hat{\mathbf{R}}}^{(\pm 1)}(\varepsilon, t) = \int_0^{2\pi} \mathrm{d}\varphi \int_0^{\pi} \mathrm{d}\theta_{\mathrm{ion}} |\chi(\theta_{\mathrm{ion}}, \varepsilon, t)|^2 \cos(\theta_{\mathrm{ion}}) \sin(\theta_{\mathrm{ion}}),$$

yielding $\langle \cos\theta_{\mathrm{ion}} \rangle_{\hat{\mathbf{R}}}^{(\pm 1)}(\varepsilon, t) = \frac{2}{\sqrt{3}} a_0(\varepsilon, t) a_1(\varepsilon, t)$. The relative number of molecules pointing forward simply corresponds to the square modulus of $\chi(\theta_{\mathrm{ion}}, \varepsilon, t)$ integrated over the forward hemisphere, in which $\theta_{\mathrm{ion}} \in \left[ 0, \frac{\pi}{2} \right]$:

$$N_+^{(\pm 1)}(\varepsilon, t) = \int_0^{2\pi} \mathrm{d}\varphi \int_0^{\pi/2} \mathrm{d}\theta_{\mathrm{ion}} |\chi(\theta_{\mathrm{ion}}, \varepsilon, t)|^2 \sin(\theta_{\mathrm{ion}})$$
$$= \frac{1}{2} + \frac{\sqrt{3}}{2} a_0(\varepsilon, t) a_1(\varepsilon, t)$$
$$= \frac{1}{2} + \frac{3}{4} \langle \cos\theta_{\mathrm{ion}} \rangle_{\hat{\mathbf{R}}}^{(\pm 1)}(\varepsilon, t).$$

Clearly, $N_-^{(\pm 1)}(\varepsilon, t) = 1 - N_+^{(\pm 1)}(\varepsilon, t)$. The explicit expression of $\langle \cos\theta_{\mathrm{ion}} \rangle_{\hat{\mathbf{R}}}^{(\pm 1)}(\varepsilon, t)$ is detailed in Supplementary Information section 4. Once $N_+^{(\pm 1)}(\varepsilon, t)$ is known, the FBFA defined in equation (7) directly follows. The dependence of the FBFA on $\varepsilon$ is illustrated in Supplementary Fig. 12.

## Data availability
Source data are provided with this paper.

## Code availability
The code used for the simulations contained in this study is available from the corresponding authors on reasonable request.

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

**Acknowledgements** We thank F. Remacle for fruitful discussions and her constructive feedback and K. Pikull for the excellent technical support. We acknowledge financial support from the European Research Council under the ERC-2014-StG STARLIGHT (grant no. 637756), ERC SoftMeter no. 101076500, ERC ULISSES no. 101054696 and the European Union's Horizon 2020 research and innovation programme no. 682978 – EXCITERS. Views and opinions expressed are however those of the author(s) only and do not necessarily reflect those of the European Union or the European Research Council Executive Agency. Neither the European Union nor the granting authority can be held responsible for them. We also acknowledge funding from the Cluster of Excellence 'CUI: Advanced Imaging of Matter' of the Deutsche Forschungsgemeinschaft (DFG) – EXC 2056 – project ID 390715994 and the DFG – SFB-925 – project ID 170620586. V.W. acknowledges support from the Partnership for Innovation, Education and Research (PIER) (PIF-2021-03). D.A. acknowledges support from the Royal Society (URF\R1\201333). A.T. acknowledges support from the Helmholtz Association under the Helmholtz Young Investigator Group VH-NG-1603.

**Author contributions** V.W., E.B., V.B., Y.M., B.P. and F.C. conceived the experiment. V.W., E.B., E.P.M., L.C., S.R., K.S. and A.T. performed the experiments. V.W., E.B. and Y.M. carried out the data analysis. M.-C.H., N.B.A. and B.P. calculated the molecular and electronic properties of methyl lactate. B.P. performed the MP-PECD and molecular orientation calculations. M.-C.H. performed the classical trajectory simulations. A.F.O., D.A. and O.S. identified and developed the concept of enantiosensitive molecular orientation. V.W., Y.M., B.P. and F.C. drafted the manuscript. All authors contributed to the discussion of the results and the editing of the manuscript.

**Funding** Open access funding provided by Deutsches Elektronen-Synchrotron (DESY).

**Competing interests** The authors declare no competing interests.

**Additional information**
**Correspondence and requests for materials** should be addressed to Vincent Wanie, Bernard Pons or Francesca Calegari.

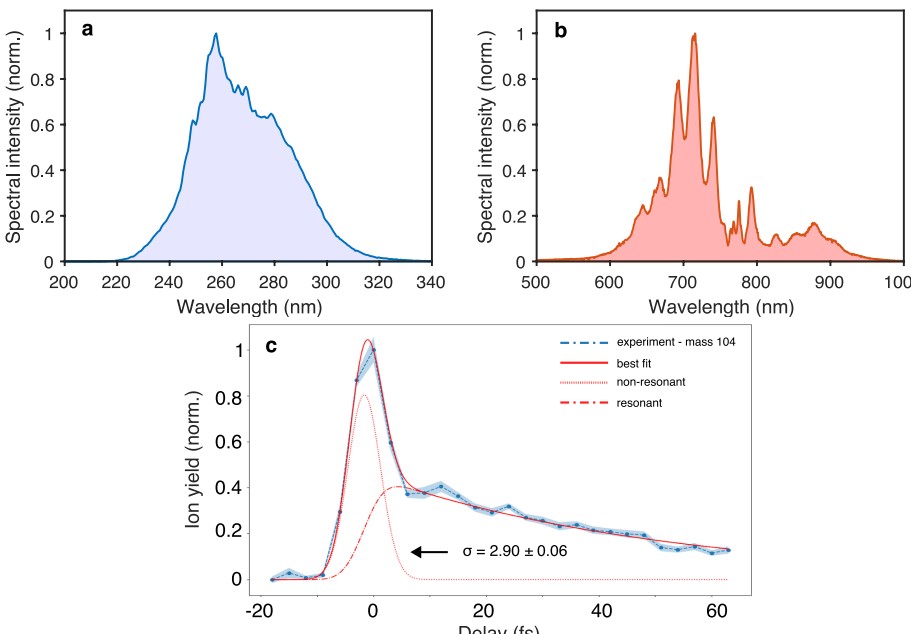

**Extended Data Fig. 1 | Laser pulses properties. a**, UV-pump spectrum. **b**, NIR-probe spectrum. **c**, Time-dependent parent ion yield. The temporal resolution of the experiment was obtained by a global fit including selected ion masses acquired simultaneously with the photoelectron spectra. The non-resonant contribution (dotted orange line) corresponds to the cross-correlation of the pump and probe pulses, from which we retrieve an instrument response function of 2.90 ± 0.06 fs. The shaded area shows the standard error of the mean obtained from five measurements.

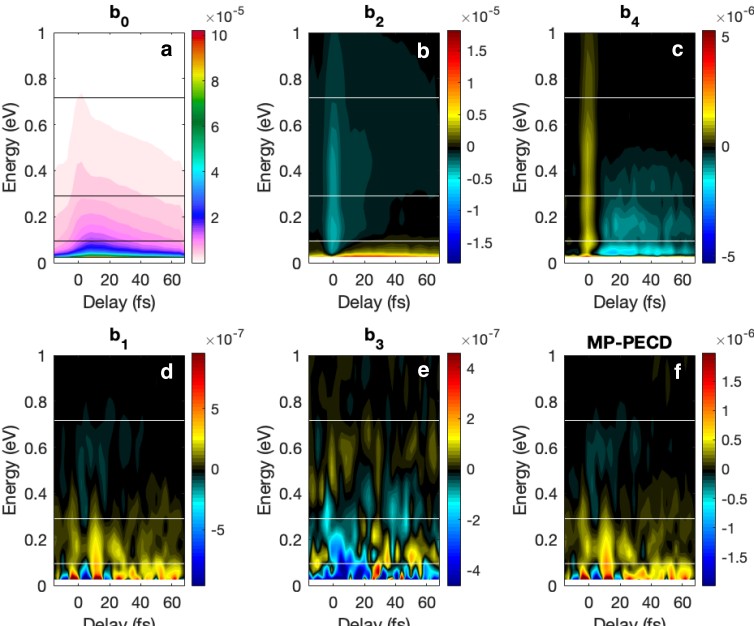

**Extended Data Fig. 2 | Decomposition of the PAD and PECD images.** Shown are the unnormalized Legendre amplitude coefficients, which are even for PAD images (**a**–**c**) and odd for PECD images (**d**,**e**), as a function of the pump–probe time delay and photoelectron kinetic energy for (*S*)-methyl lactate. Electrons with kinetic energy below 25 meV are not considered. This decomposition allows to retrieve a MP-PECD with its energy and time dependency shown in **f**. The horizontal lines delimit the three energy regions discussed in the main text: 25–100 meV, 100–300 meV and 300–720 meV.

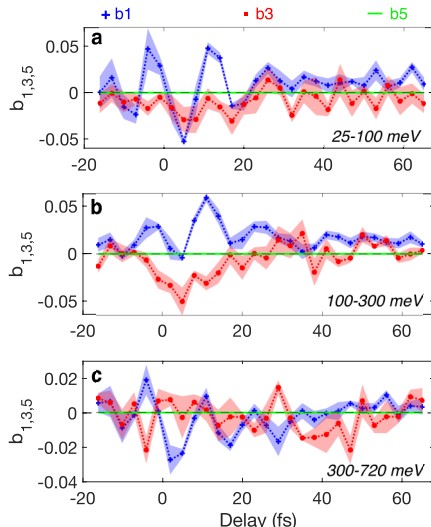

**Extended Data Fig. 3 | Energy-resolved time dependence of the odd coefficients.** Shown are the time dependencies of the normalized $b_1$, $b_3$ and $b_5$ coefficients for ($S$)-methyl lactate in the ranges 25–100 meV (**a**), 100–300 meV (**b**) and 300–720 meV (**c**). For each energy range, within the cylindrical symmetry approximation of the photoelectron momentum, the dominant modulation is observed for the $b_1$ coefficient, which thus governs the signal of the MP-PECD (Fig. 2). The contribution from $b_5$ is negligible.

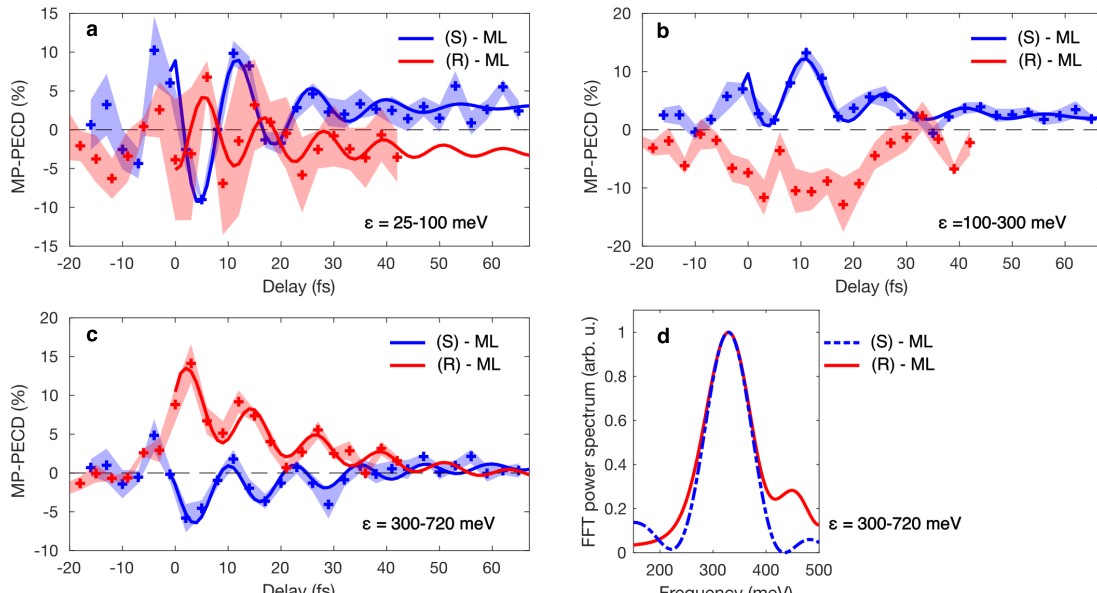

**Extended Data Fig. 4 | Enantiomer comparison. a–c**, The MP-PECD for the photoelectron kinetic energy regions defined in Fig. 2a for both (*S*)-methyl lactate (blue curve) and (*R*)-methyl lactate (red curve). **d**, The fast Fourier transform (FFT) power spectra for $\varepsilon = 300$–$720$ meV.

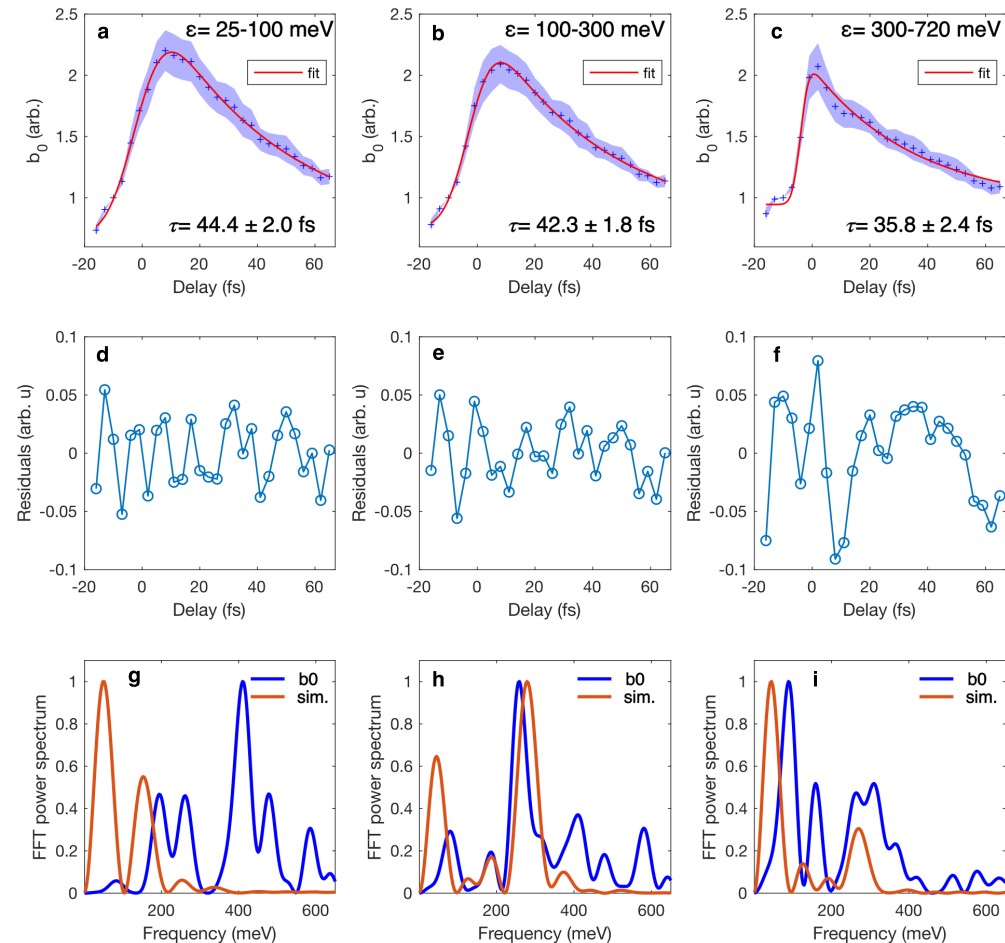

**Extended Data Fig. 5 | Analysis of the total photoelectron yield.** Time evolution of the $b_0$ coefficient for the three $\varepsilon$ regions defined in Fig. 2a: 25–100 meV (**a**), 100–300 meV (**b**) and 300–720 meV (**c**). The standard error over five measurements is shown by the shaded area. The red curve shows a fit of the experimental data and the residuals obtained by subtracting the fit from the experimental data are shown in **d**–**f**. The power spectrum from a Fourier analysis is shown in **g**–**i** and is compared with the theoretical model. FFT, fast Fourier transform.