## [Peer Review File · Nature]

Manuscript Title: Capturing electron-driven chiral dynamics in UV-excited molecules

Reviewer Comments & Author Rebuttals

Reviewer Reports on the Initial Version:

Referee #1 (Remarks to the Author):

In this manuscript, the authors present results as to the measurement of time-resolved photoelectron circular dichroism in methyl lactate coupled with theoretical calculations of the same. This is an extremely interesting and timely experiment, although it is not likely to influence thinking in the field. Additionally, the evidence for the conclusions presented is not sufficiently strong for publication in Nature.

The fundamental results of both the experiment and the theory are valid. They are both impactful, but the authors have stretched their argument in order to attract greater attention. The title of the manuscript is "Ultrafast chiroptical switching...", and the authors put chiroptical switching in the first sentence of the abstract, and continue to say "the populated electronic coherences can be used for chiroptical switching...". This statement is repeated elsewhere as well, and seems to be a primary justification for the work. I believe this justification is misguided and misleading. Molecular switches, including those which operate via chiral mechanisms and those cited in references 6 and 7 of the manuscript, involve a molecule with a bistability. There should be two stable (steady) states, which interconvert via external stimulus. The authors present no such system, nor any system which may be used in a similar manner. Instead, they present a system that is engaged in coherent dynamics, the beating between two Rydberg states, and then probed via PECD. This coherent dynamics is not a switching mechanism, as it is not bistable but rather a predictable and understandable dynamical process. This dynamics is fundamentally interesting and worth reporting on, but the analogy to molecular switches is flawed. Additional support for switching in terms of exposition and references to similar processes would be necessary to maintain the analogy, and neither was provided.

Interpreting the work, instead, as a study of the ultrafast dynamics of a chiral molecule, I find it an excellent addition to the field but falling short of the standards of Nature. The use of linearly polarized UV and circularly polarized IR pulses is reminiscent of previous work from the authors, including some of the referenced work and <https://doi.org/10.1039/C6FD00113K>. The authors should include a discussion of how the current experiment differs from their previous works. The conclusions are largely valid with some detailed questions I will discuss subsequently, and the theory is impressive in its agreements with the

experiment, but even then I question whether the MP-PECD observable is the most appropriate to compare. This observable boils down the asymmetry to a single quantity, and for that reason it is useful, but for the same reason it can hide theory-experiment disagreement and obfuscate some of the dynamics that might be occurring. Additionally, the relatively low kinetic energies of the electrons where much of the analysis is concerned leads to both experimental and theoretical difficulties. The experimental details are possible to overcome, though it is difficult to discern at the level of detail of the manuscript what efforts were made to abate some of these issues. With the theory, low kinetic energies can lead to effects from both the tails of the Coulomb potentials as well as the nuclear cusps that can be difficult to get right at the level both of DFT and of Gaussian basis sets. It is unclear whether the Xalpha method can overcome these shortcomings, and it is unclear from the manuscript whether or not they were considered. It is somewhat unfair to ask that the authors of the manuscript answer this question, I concede, but they make me hesitate to say that the authors have sufficiently strong evidence to warrant a Nature publication. Similarly, questions about the accuracy of TPAs calculated in Gaussian basis sets for such high-lying states arise.

I will now list some comments I had on the manuscript. Some repeat my larger concerns detailed above, and some are additional minor concerns:

1. In lines 64-69, it is suggested that PECD is more sensitive than transient absorption and other spectroscopies. I do not know this to be true. While PECD is stronger than optical-magnetic chiral observables, this does not as I understand extend to the other spectroscopies mentioned. Lines 115-119 give additional details that again seem similar to previous work.
2. Lines 90-92 detail the experiment, it is unclear from this description what is unique about the current experiment compared to previous work.
3. On lines 120-123, the authors discuss the symmetry of the PAD for chiral vs achiral molecules. While what they say is certainly true for one-photon processes, it is not true for multi-photon processes or for initially excited molecules. This is a point which the literature for chiral systems repeatedly ignores. For an initial superposition (such as the one prepared) which is "electronically chiral" in that it does not have parity symmetry, one can expect the same asymmetries in the PAD as for chiral molecules.
4. In Fig. S4 of the supplement, there seem to be significant differences between R and S enantiomers. These should be discussed in the main manuscript. Some may be due to differences in statistics or related experimental features. But they are stark enough that an explanation is warranted. The FFT of the enantiomers signals show different sub-peaks, for instance.
5. Line 135, "it is striking that the photoelectron emission asymmetry reverses...". This is one area where I am concerned that the conclusions are based on the MP-PECD observable and not necessarily backed up by the more detailed PECDs. Is this a true inversion/reversal of direction? The PECDs do not

seem to invert, rather the MP-PECD is modulated (towards zero mostly, not even in the opposite sign). This means it could just be some averaging effect that looks like a reversal but is a more complicated behavior of the wave packet. Additional analysis of the full PECDs would be useful to learn more about what is actually happening.

6. In the manuscript (not necessarily true of the SI), the molecular dynamics trajectories (lines 190-192) seem to be an afterthought. It is unclear what conclusions are being drawn from them in relation to the experiment. Some of the details, including how many trajectories were run, also belong in the manuscript rather than the SI. The main conclusion of the dynamics (which are suspect due to being run on only the cation) seems to be that the experimentally observed decoherence must be due to non-adiabatic effects. This conclusion may be correct, but it is not strongly supported and furthermore it is unclear what is learned about the experiment here.

7. Line 245-246, "...chiral character of the Rydberg states contributes to chiroptical switching." I disagree with this conclusion (not only because of the switching premise). The nature of the states in the initial superposition are not discussed significantly in the manuscript. Details about the calculations are not given that support this conclusion. They may be there and obvious to the authors but as a reader, I would have preferred to read more about how the authors understood the nature of the Rydberg states by comparing the theory to the experiment.

In my opinion, there is no discernible reason why this work warrants the visibility of Nature. While the experiment and theory are both interesting and valid, neither moves the field forward nor presents a result requiring high visibility.

Referee #2 (Remarks to the Author):

In the manuscript "Ultrafast chiroptical switching in UV-excited molecules" by Wanie et al., the authors report a combined experimental and theoretical study of electron dynamics in neutral methyl-lactate, a chiral molecule. The authors have used a traditional pump/probe approach to (1) create a coherent electron wave-packet in the bound states (Rydberg states) using a few-femtosecond UV pulse and to (2) probe its dynamics with a time-delayed circularly-polarized IR field, through a photoionization process. The electron dynamics is revealed by the measurement of the angle-resolved photoelectron energy distributions as the time delay between the pump and probe pulses is varied. From the forward-backward asymmetry along the light propagation observed in the photoelectron angular distributions, generally known as PhotoElectron Circular Dichroism (PECD), the authors reports an oscillatory change of direction of the photoelectron emission as the time delay between the pump and probe is varied on a femtosecond timescale. Theoretical calculations performed within the frozen-nuclei approximation link such an oscillatory pattern to a quantum interference process between Rydberg states coherently populated by the pump pulse. In addition,

calculations show that the launched electron wave-packet preserves its coherence over tens of femtoseconds.

Overall, this is a very interesting paper which clearly demonstrates that tracking electron dynamics in large molecular systems is slowly coming of age. I have some comments about the authors' claims, though, but I do think publication of this paper would provide interesting results to the ultra-fast community.

Comments:

1-In the introduction, the authors claims that: "Our findings establish a general method to investigate electron dynamics in a variety of chiral systems with high sensitivity and pave the way to a new scheme for enantio-sensitive charge-directed reactivity in neutral chiral molecules." I don't feel that this statement reflects correctly the results of this study. It is true that the present study offers very interesting results about electron dynamics in neutral chiral molecules. But, I would argue that the method used in the experiment is quite traditional. The use of femtosecond pump/probe pulses to reveal electron and/or nuclear dynamics in molecular systems has already been reported many times in the literature on a vast number of different systems. I would then suggest to reword this statement to better reflects the originality of the study.

2-In the second paragraph of the introduction (line 43-44), the authors claim that "electron dynamics in biochemically-relevant neutral molecules in general remains to be demonstrated". Shortly after (line 50-51), they also claim that "Seminal pump-probe experiments using attosecond light pulses have revealed electron-driven charge migration in biomolecules". That seems somewhat contradicting claims. I would recommend to clarify this point.

3-In the third paragraph of the introduction (line 54-63), the author claims that "Two important technological challenges must be addressed. First, the pump pulse must have well defined characteristics: (i) a photon energy below the ionization threshold, (ii) a broadband energy spectrum to trigger electron motion among multiple electronic states and (iii) a time duration that provides a prompt excitation before any nuclear motion can take place, together with sufficient temporal resolution" Even though all of these characteristics are indeed required for this study, I would not describe this as an important technological challenge, as the generation of a third harmonic of an IR femtosecond pulse is quite trivial.

4-Line 112-113: "which captures an inversion of the photoelectron forward-backward asymmetry in about 7 fs." I would suggest to be more specific with both the description of Fig 1.b and the figure itself as it is not directly obvious to see this oscillation in the asymmetry parameter. Part of the PECD image does not seem to oscillate at all from 11 to 26fs. The magnitude does change but not the direction of emission.

5-Regarding the theoretical model, It is true that a full quantum calculation is probably not feasible, and that classical molecular dynamics are probably the only tool available. But, I feel that arguments justifying how classical molecular dynamics are appropriate to extract the physics for this particular system are missing in the main text. Authors should probably further comment on the use of this tool.

6-Lastly, the authors should probably provide information regarding the state of the molecule before irradiation. As described in the experimental method section, the molecules are generated via evaporation. Is there any possibility to have some molecules already populated in Rydberg states before irradiation? If yes, how this pre-population would affect the findings? Following the same idea, the more likely the evaporation process would lead to some excited vibrational states of the molecules. Could the author quantify that and comment on how this would affect the electron dynamics?

Author Rebuttals to Initial Comments:

We are grateful to the referees for their invaluable feedback, with very insightful comments and helpful suggestions. In response to the main criticisms about the novelty and potential impact of our work, we have expanded our model to provide further clarity (depicted now in Figure 4d-f). Additionally, we have introduced substantial revisions to enhance the manuscript's presentation. We sincerely hope that these efforts will meet the referees' expectations. Herein, we address specific comments intended for the referees as follows:

Referee #1 (Remarks to the Author):

In this manuscript, the authors present results as to the measurement of time-resolved photoelectron circular dichroism in methyl lactate coupled with theoretical calculations of the same. This is an extremely interesting and timely experiment, although it is not likely to influence thinking in the field. Additionally, the evidence for the conclusions presented is not sufficiently strong for publication in Nature.

The fundamental results of both the experiment and the theory are valid. They are both impactful, but the authors have stretched their argument in order to attract greater attention. The title of the manuscript is "Ultrafast chiroptical switching...", and the authors put chiroptical switching in the first sentence of the abstract, and continue to say "the populated electronic coherences can be used for chiroptical switching...". This statement is repeated elsewhere as well, and seems to be a primary justification for the work. I believe this justification is misguided and misleading. Molecular switches, including those which operate via chiral mechanisms and those cited in references 6 and 7 of the manuscript, involve a molecule with a bistability. There should be two stable (steady) states, which interconvert via external stimulus. The authors present no such system, nor any system which may be used in a similar manner. Instead, they present a system that is engaged in coherent dynamics, the beating between two Rydberg states, and then probed via PECD. This coherent dynamics is not a switching mechanism, as it is not bistable but rather a predictable and understandable dynamical process. This dynamics is fundamentally interesting and worth reporting on, but the analogy to molecular switches is flawed. Additional support for switching in terms of exposition and references to similar processes would be necessary to maintain the analogy, and neither was provided.

We are pleased that the referee acknowledges the validity of our experimental and theoretical results as well as their impact. We also understand and agree that using the term 'chiroptical switching' could be misleading to a reader. PECD is defined as a chiroptical effect and in our work the photoelectron emission direction that characterizes it can switch as a function of the time delay due to the coherent dynamics we initiate upon UV-excitation, hence 'chiroptical switching'. However, as pointed out by the referee this is very different from what is generally defined for molecular switches where thermodynamically stable states can be selected reversibly. For this reason and following the referee's recommendation, we changed the title of the manuscript and removed any analogy to chiroptical switching from the main text in order to keep the reader's attention to the electron-driven chiral dynamics reported (see also the detailed answer below).

Interpreting the work, instead, as a study of the ultrafast dynamics of a chiral molecule, I find it an excellent addition to the field but falling short of the standards of Nature. The use of linearly polarized UV and circularly polarized IR pulses is reminiscent of previous work from the authors, including some of the referenced work and <https://doi.org/10.1039/C6FD00113K>. The authors should include a discussion of how the current experiment differs from their previous works.

In previous studies [both in *Faraday Discuss.* 194, 325 (2016) and *J. Phys. Chem. Lett.* 7, 4514 (2016)], the bandwidth of the UV-pump was not sufficiently large to induce electronic coherences. As a main consequence, the wavepacket generated by the pump pulse was primarily nuclear in nature rather than electronic, thus lacking the creation of an electronic chiral current. The innovative part of the present work is the chiroptical modulation stemming from the evolving electronic coherences, as presented in this manuscript and investigated through the time-dependent PECD, and it represents a truly novel finding. This observation became feasible due to the broad (>500 meV) pump pulse bandwidth and the unprecedented 2.9 fs temporal resolution employed in this investigation. As explained in the detailed answer below, we have now modified the manuscript in order to better highlight this crucial aspect.

The conclusions are largely valid with some detailed questions I will discuss subsequently, and the theory is impressive in its agreements with the experiment, but even then I question whether the MP-PECD observable is the most appropriate to compare. This observable boils down the asymmetry to a single quantity, and for that reason it is useful, but for the same reason it can hide theory-experiment disagreement and obfuscate some of the dynamics that might be occurring.

The PECD/MP-PECD signal that we employ is a photoelectron energy and angle- resolved quantity (for a given pump-probe delay). PECD/MP-PECD stand as a background-free observable renowned for its high sensitivity to chirality. In the impressive body of work by the pioneers in the field such as I. Powis, L. Nahon, T. Baumert and others, it has been shown that PECD is molecular-orbital dependent (2001-2006), sensitive to conformers (2008), to chemical substitution (2008), to chiral dimers (2010), to chiral clusters (2011) and to chiral vibrational dynamics (2013), while the TR-PECD adds temporal resolution. PECD/MP-PECD combine the power of energy and angle-resolved photoelectron spectroscopy with the outstanding chiral sensitivity (up to 40%), about 3 orders of magnitude higher than in standard methods (e.g. circular dichroism and optical rotation). It has been recently applied to chiral liquids (in the group of B. Winter) and is being adapted to applications as a new tool for chiral mass spectrometry by the spinoff company MASSPEC led by M. Janssen. Despite this impressive progress and outstanding versatility, the sensitivity of PECD/MP-PECD to ultrafast chiral electron dynamics has remained elusive up until this work. Our work bridges this gap, opening the way to imaging and controlling ultrafast coupled electronic and nuclear dynamics – an outstanding challenge in understanding chiral interactions at the level of electrons, rather than in the language of thermodynamics and kinetics, as it has been until now.

In practice, considering the laser's repetition rate (1 kHz), we can hardly conceive an alternative signal capable of manifesting, including energy resolution (ϵ -dependency that reveals the electronic states contributing to the chiral current), even a fleeting glimpse into the electronic current initiated by the pump pulse. Notably, this current originates from the

inherent *structural* chirality encoded in the electronic density of excited Rydberg states, and is conditioned by the presence of electronic coherences. As a result, a photoelectron-based detection emerges as the most direct avenue to delve into this intriguing electronic chiral phenomenon.

Additionally, the relatively low kinetic energies of the electrons where much of the analysis is concerned leads to both experimental and theoretical difficulties. The experimental details are possible to overcome, though it is difficult to discern at the level of detail of the manuscript what efforts were made to abate some of these issues. With the theory, low kinetic energies can lead to effects from both the tails of the Coulomb potentials as well as the nuclear cusps that can be difficult to get right at the level both of DFT and of Gaussian basis sets. It is unclear whether the Xalpha method can overcome these shortcomings, and it is unclear from the manuscript whether or not they were considered. It is somewhat unfair to ask that the authors of the manuscript answer this question, I concede, but they make me hesitate to say that the authors have sufficiently strong evidence to warrant a Nature publication. Similarly, questions about the accuracy of TPAs calculated in Gaussian basis sets for such high-lying states arise.

We thank the referee for these remarks and clarify all of them in the point-by-point reply below.

Experimental two-photon absorption spectra, from which we could gauge directly the reliability of our computed TPAs, do not exist for methyl-lactate. Therefore, we focused on the one-photon absorption spectrum which has been measured at the DESIRS beamline of SOLEIL synchrotron. In the review supporting file #1 joint to this document, we compare these measurements to simulations resulting from the same TDDFT calculations as those employed for the computations of the TPAs. This figure will be part of a forthcoming publication. The exceptional agreement between experiment and theory clearly demonstrates that the Gaussian basis set and exchange/correlation functional used in our calculations are suitable to describe the Rydberg electronic structure of methyl-lactate, from which we can subsequently calculate reliable TPAs.

Concerning the asymptotic Coulomb behavior of the potential, we have indeed considered and included the effect of the Coulomb tail in the computation of both bound and continuum states. The so-called Latter approach, which reproduced the $-1/r$ tail of the potential, has been introduced in the Xalpha calculations, as detailed after equation (11) in the SI. For the bound states, we employed the LCBLYP functional in DFT (and TDDFT) calculations. This functional has been explicitly built in order to allow for the Coulomb behavior at large distances, and behaves better than the conventional CAMB3LYP functional, as shown in the following Table where the energies of some bound states of Li, obtained by means of TDDFT calculations involving different functionals, are compared to the reference data of NIST:

state	NIST (eV)	BLYP (eV)	B3LYP (eV)	CAMB3LYP (eV)	LC-BLYP (eV)
2p	1,848	2,007	1,980	1,944	1,935
3s	3,373	2,958	3,126	3,331	3,407
3p	3,834	3,044	3,351	3,684	3,824
4s	4,341	3,042	3,449	4,039	4,351
4p	4,521	3,060	3,499	4,156	4,505
5s	4,749	3,071	3,514	4,311	4,753
5p	4,837	3,100	3,533	4,359	4,820
6s	4,958	3,130	3,550	4,426	4,958
6p	5,008	3,193	3,579	4,453	4,991
7s	5,079	3,251	3,635	4,491	5,076

The referee will observe that TDDFT-LC-BLYP calculations, using our large-scale underlying Gaussian basis, provides very satisfactory results and is essential to produce the accurate 1-photon absorption spectrum shown in the review supporting file #1.

With respect to nuclear cusps, the Dunning-Hay Gaussian basis is basically a double-zeta basis. While we could have optimized the description of nuclear cusps by using a higher-zeta basis, we instead added to the Dunning-Hay basis a large-scale even-tempered set of Gaussian functions, including high angular symmetries (up to g-states). Such high l-states improve the description of the cusps, well beyond the primary double-zeta framework.

As for the MP-PECD calculations for the photoelectrons with a kinetic energy below 100 meV, shown in Fig. S9 of the SI, the theory does not succeed in reproducing the experimental results. So close to the ionization threshold, electron-electron correlations are generally too important and cannot be reproduced in the statistical Xalpha framework. Including correlations in the description of ionization of large molecules is beyond the present computing/theoretical capabilities. Our approach behaves better at higher electron energies as illustrated in fig. 3 of the main manuscript.

I will now list some comments I had on the manuscript. Some repeat my larger concerns detailed above, and some are additional minor concerns:

1. In lines 64-69, it is suggested that PECD is more sensitive than transient absorption and other spectroscopies. I do not know this to be true. While PECD is stronger than optical magnetic chiral observables, this does not as I understand extend to the other spectroscopies mentioned. Lines 115-119 give additional details that again seem similar to previous work.

We changed lines 64-69 from the initial manuscript to the following: "All these characteristics would provide the tools to harness temporal resolutions previously unattained in pump-probe spectroscopic techniques that are highly sensitive to chirality, such as time-resolved photoelectron circular dichroism (TR-PECD)⁸⁻¹²."

Lines 115-119 have been moved to the Methods section of the manuscript since our intention is to provide relevant information about the experimental approach and the data analysis. If adding those details in the main text gave the wrong impression that we wanted to introduce TR-PECD as a new approach, we apologize and we now remove this ambiguity by providing several references to the TR-PECD technique. The new manuscript makes it clear that we employ a known and robust procedure that was also used in some of the author's previous works. The major differences with these previous works are elaborated in the point 2 below.

2. Lines 90-92 detail the experiment, it is unclear from this description what is unique about the current experiment compared to previous work.

We rephrased lines 90-92 to: "First, a linearly polarized UV pulse promptly launches a coherent electronic wavepacket just below the ionization threshold in the bio-relevant molecule via a two-photon transition. Then, a time-delayed circularly polarized near-infrared (NIR) probe triggers ionization from the transient wavepacket, providing an exceptional instrument response function of 2.90 ± 0.06 fs."

It is now clear that the exceptional time resolution of our experiment is a unique and key element compared to all previous works. Our ultrashort UV pump pulses are pivotal to observe the electronic beatings between Rydberg states and are employed for the first time not only in a TR-PECD experimental scheme, but more generally for any type of time-resolved measurements of ultrafast dynamics of matter, as we also detail in the reply to the point 3 of the second referee. The time resolution of 2.90 ± 0.06 fs is obtained from the figure below, showing the time-dependent yield of the parent ion with mass 104 a.u. (blue dotted line) that was collected simultaneously with the photoelectrons signal. The fit (red line) contains contributions from the both resonant (dashed red) and non-resonant (dotted) ionization pathways, the latter providing the instrument response function for the experiment. This figure was added to the supplement.

This is the new Fig. S1(c) in the revised submission.

Furthermore, our previous work on TR-PECD mainly concerned the relaxation dynamics of chiral molecules excited into a single electronic state, out of the scope of coherent electron dynamics in chiral systems. Other groups also employed TR-PECD to investigate photodissociation in chiral systems that involves vibrational dynamics, again subsequent to excitation onto a single electronic state [Communications Chemistry 4, 119 (2021), Science Advances 8, eabq2811 (2022)]. The first and only work mentioning coherent electron dynamics in neutral chiral molecules is our study on photoexcitation circular dichroism [Nature Physics volume 14, 484 (2018)] and stemmed from calculations rather than from experiments: the limited resolution of about 100 fs that we had in this experiment did not allow us to detect the chiral electron dynamics experimentally.

All previous experiments employed conventional UV sources with only ~100 fs duration that prohibited the observation of ultrafast coherent electron dynamics. Therefore, our present investigation is the first one that enters the few-femtosecond TR-PECD regime, owing to the unprecedented temporal resolution provided by our UV light source.

3. On lines 120-123, the authors discuss the symmetry of the PAD for chiral vs achiral molecules. While what they say is certainly true for one-photon processes, it is not true for multi-photon processes or for initially excited molecules. This is a point which the literature for chiral systems repeatedly ignores. For an initial superposition (such as the one prepared) which is "electronically chiral" in that it does not have parity symmetry, one can expect the same asymmetries in the PAD as for chiral molecules.

The remark from the referee is correct when the sample consists of aligned or oriented molecules. However, we deal with a set of *randomly oriented molecules* so that the forward/backward asymmetry with respect to the light propagation axis cannot show up for achiral molecules whatever is the initial excitation step. To clarify this point, we modified the original line 120-123 (now found in the Methods section) to: "In the case of a sample of randomly oriented achiral molecules, the PAD is symmetric with respect to the light propagation axis so that the $S^{(n)}(\epsilon, \theta, t)$ expansion is restricted to even n 's."

4. In Fig. S4 of the supplement, there seem to be significant differences between R and S enantiomers. These should be discussed in the main manuscript. Some may be due to differences in statistics or related experimental features. But they are stark enough that an explanation is warranted. The FFT of the enantiomers signals show different sub-peaks, for instance.

The data for each enantiomer were acquired during two successive days. While this ensured nearly identical experimental conditions, the total signal collected for (S)-ML was 75% higher than for (R)-ML, leading to higher statistics. Furthermore, the enantiomeric excess of (S)-ML and (R)-ML was respectively 97% and 96%. Such a small difference can already impact the measurements due to the high sensitivity of the PECD effect [Nat. Comm. 9, 1 (2018)]. Because of these two factors, (S)-ML provides more exact results. It is important to note however that these minor differences in the experimental conditions do not affect our main observations and that the comparison and agreement between the two enantiomers is satisfactory: as expected, the sign of the PECD signal periodically reverses for low kinetic energy electrons and for the other two energy regions the oscillations have different sign depending on the

enantiomer. As recommended by the referee, we added the following explanation to the main manuscript: "The results are shown for (*S*)-ML and a mirroring symmetric measurement in (*R*)-ML clearly confirms the chiral character of the Rydberg-induced dynamics, with minor discrepancies due to slightly lower enantiopurity and statistics (see Fig. S5 of the revised Supplementary Information, SI)."

The FFT analysis for both b_1 and the MP-PECD always shows a clear dominant peak with sub-peaks that are within the noise level when compared to the normalized signal (i.e., generally less than 20% of the main peak), beside a very slow frequency component with corresponding period >40 fs (not shown). It is only in the case of b_0 (Fig. S4 in the SI) that several sub-peaks appear and make the analysis and interpretation more difficult. This shows how dealing with the odd coefficients - that are background free - rather than the even ones eases the analysis and interpretation in the photoelectron spectroscopy of ultrafast dynamics of chiral molecules.

5. Line 135, "it is striking that the photoelectron emission asymmetry reverses...". This is one area where I am concerned that the conclusions are based on the MP-PECD observable and not necessarily backed up by the more detailed PECDs. Is this a true inversion/reversal of direction? The PECDs do not seem to invert, rather the MP-PECD is modulated (towards zero mostly, not even in the opposite sign). This means it could just be some averaging effect that looks like a reversal but is a more complicated behavior of the wave packet. Additional analysis of the full PECDs would be useful to learn more about what is actually happening.

While the integrated MP-PECD is the most appropriate quantity to describe the whole forward/backward asymmetry, we understand the concern and agree with the referee that an additional analysis of each individual odd term is required for the sake of completeness of our interpretation. We therefore added Fig. 2 (a) and (b) in the main text (displayed below) comparing the time and energy-resolved maps of the MP-PECD and b_1 parameter, respectively. Clearly the similarity between the maps demonstrates that the b_1 coefficient is responsible for all the main features observed in the MP-PECD. This is consistent with a true inversion of photoelectron emission direction at kinetic energies below 100 meV shown in (c). Importantly, this effect is also reproduced experimentally for the other enantiomer (Fig. S5 of the SI).

Figure 2: **Energy-resolved analysis.** Temporal evolution of the unnormalized MP-PECD in (*S*)-methyl-lactate (a) and corresponding b_1 coefficient (b). The white lines identify three different kinetic energy ranges of photoelectrons: 25-100 meV (c), 100-300 meV (d) and 300-720 meV (e). The standard error of the mean over 5 measurements is shown by the shaded areas. The solid blue lines show the fit of the oscillations from $t = 0$ fs (see the corresponding Fourier analysis in Fig. 3c,e). The change of sign in (c) identifies a reversal of the photoelectron emission direction in the laboratory frame.

This is the new Fig. 2 in the revised submission.

We also added Fig. S3 to the supplement, showing the individual contributions $b_{1,3,5}(t)$ for each energy region described in the main text. The figure is shown below. Again, the oscillatory behavior mostly originates from the b_1 coefficient and it provides information on the isotropic part of the asymmetry inherent to the interaction [Phys. Chem. Chem. Phys. 23 (2021)]. b_3 provides information on the anisotropy of excitation created by the pump pulse while b_5 is several orders of magnitude smaller and does not contribute to the MP-PECD. The fact that all the experimental features are present in both the b_1 coefficient and the MP-PECD illustrates the robustness of our interpretation and the truthfulness of the physical forward/backward asymmetry that is evident between 25-100 meV.

This is the new fig. S3 in the revised version.

Thanks to the referee, we also realized that the angularly-resolved PECD plots of Fig. 1(b) are not as intuitive as we thought. In the revised figure (now Fig. 1c), we include white dashed circles to discriminate the first two energy ranges that we consider later on in Fig. 2, so that the inversion of PECD becomes clear visually. Furthermore, the updated Fig. 1b was modified and a simplistic representation of the inversion effect is illustrated in order to prepare the reader to identify it in Fig. 1c. The revised figure is shown below.

Figure 1 : Light-induced chiral dynamics of methyl-lactate. (a) A few-femtosecond linearly polarized UV pulse excites an ensemble of randomly oriented chiral molecules, creating an electronic wavepacket of Rydberg states via 2-photon absorption. The dynamics is probed via 1-photon ionization by a time-delayed circularly polarized NIR pulse. The probing step leads to the ejection of photoelectrons along the light propagation axis defined along the z direction and the resulting angular distribution is recorded by a velocity map imaging spectrometer. (b) The red and blue structure shows the temporal evolution of the coherent electron density in the excited neutral molecule: the chiral evolution of the photoexcited Rydberg wavepacket leads to a reversal of the 3D photoelectron angular distribution at two distinct time delays t and $t+\Delta t$, captured by the measurements. (c) For each time delay, an image is recorded for both left and right circular polarization of the probe pulse. The differential image PECD(ϵ,θ,t) defined in the main text is shown for time delays of 5, 11, 17 and 26 fs for photoelectrons with kinetic energies from 25 to 300 meV along the radial coordinate. The white circles identify the photoelectrons below 100 meV which experience an ultrafast reversal of their emission direction.

This is the new Fig. 1 of the revised version.

It is now easier to observe the overall periodicity within the temporal evolution of the PECD, which is inherent to quantum beatings in both energy ranges, even if the interplay of various beatings (beyond the simple two-state example considered at the end of our manuscript for illustration purposes) slightly blurs the whole picture. It has indeed been shown that the angular shape of PECD is sensitive to the intermediate Rydberg state reached by pump excitation [Phys. Chem. Chem. Phys. 23 (2021)]. Therefore, the occurrence of secondary quantum beatings, involving additional intermediate states, distorts the perfect periodicity of the PECD angular shape expected from a single two-state beating.

6. In the manuscript (not necessarily true of the SI), the molecular dynamics trajectories (lines 190-192) seem to be an afterthought. It is unclear what conclusions are being drawn from them in relation to the experiment. Some of the details, including how many trajectories were run, also belong in the manuscript rather than the SI. The main conclusion of the dynamics (which are suspect due to being run on only the cation) seems to be that the experimentally observed decoherence must be due to non-adiabatic effects. This conclusion may be correct, but it is not strongly supported and furthermore it is unclear what is learned about the experiment here.

As recommended, we added a section providing the details about our molecular dynamics calculations in the Methods of the main manuscript. The referee is also correct: they have been run on the ground state cation surface and not on neutral Rydberg surfaces. However, as firmly established by numerous investigations of Rydberg spectroscopies across various molecular systems [e.g. PCCP 25, 16712 (2023)], the high-lying Rydberg states generally behave nearly identically to the cation ground state to which they correlate (Fig. S11 of the SI). The identification of cationic and neutral Rydberg surfaces for nuclear dynamics is widely accepted and generally validated through the observation of very similar vibrational spectra (see, e.g., ChemPhysChem 21, 2468 (2021); PCCP 25, 16712 (2023)).

In our original manuscript, these calculations served several purposes. First, they validated the frozen-nuclei description of electron beatings monitored by roughly constant energy differences between electronic states. Secondly, they also precluded the role of purely vibrational mechanisms, related to the amplitudes and phases of vibrational wavepackets on distinct electronic states, to the decoherence observed in Fig. 2 in terms of the decrease of MP-PECD oscillations amplitude. We pragmatically concluded that decoherence occurs as a result of internal conversion, including non-adiabatic effects, which also shows up as a decrease of the photoelectron yield soon after pump excitation (Fig S4a-c of the SI).

In the revised manuscript, and as will be described in the reply to point 7, these calculations are additionally employed to illustrate another important effect that arises from the creation of electronic currents. The chiral currents generated by the coherent superposition of excited electronic states can act as an enantio-selective filter of molecular orientations in the laboratory frame upon photoionization by a circularly polarized probe [arXiv:2106.14264v3]. We address the experimental feasibility to demonstrate such effect by using our trajectory simulations.

7. Line 245-246, "...chiral character of the Rydberg states contributes to chiroptical switching." I disagree with this conclusion (not only because of the switching premise). The nature of the states in the initial superposition are not discussed significantly in the manuscript. Details about the calculations are not given that support this conclusion. They may be there and obvious to the authors but as a reader, I would have preferred to read more about how the authors understood the nature of the Rydberg states by comparing the theory to the experiment.

We apologize for the unclear discussion about the nature of the Rydberg states involved in our study. Beyond its usual spectroscopic label n , assigned in Fig. 3a and Fig. S6, we defined the 'character' of a Rydberg state using the individual MP-PECD signal that it yields upon

photoionization without any coherence with other states. This ‘character’ thus depends on the well-defined pump-probe excitation scheme, and it can serve as an experimentally reproducible label. Since our wording may have been misleading, we have accordingly modified the text related to the description of Fig. 4 (following equation 4) in order to make our purely operational labeling clear to the reader (i.e. avoiding the discussion of either the nature or character of a state).

In my opinion, there is no discernible reason why this work warrants the visibility of Nature. While the experiment and theory are both interesting and valid, neither moves the field forward nor presents a result requiring high visibility.

For the very first time, we demonstrate chiral effects in neutral molecules that are driven by coherent electron dynamics. As the dynamics of electrons holds the keys to fundamental understanding of molecular chirality and chiral interactions, we believe that our work constitutes a significant breakthrough not only for the ultrafast community, but most importantly for the broad audience working with transient chirality and developing technologies exploiting chiral properties of matter.

We understood that our initial manuscript was not sufficiently highlighting this aspect. As we now emphasize in our revised manuscript, our observations directly push the field forward by opening a route to electron-driven charge-directed reactivity in chiral molecules, and we provide a concrete example based on simulations in which the orientation of molecules undergoing dissociative photoionization can be favored depending on the rotation direction of the electron current created within a chiral molecule. We added these results as Fig. 4d-f of the main text (shown below). In (d) we show that the coherent superposition of Rydberg states leads to a chiral electron current flowing through the molecule. This current is defined in the molecular frame, so that it rotates in the laboratory frame as the molecular orientation is varied. The circularly polarized probe pulse preferentially ionizes the molecules whose orientation leads to (i) a current that co-rotates with the circularly polarized probe field (red arrow) and (ii) a rotation axis of the current that aligns with the probe propagation vector. – in Fig. 4(d), this means that molecules oriented along $\hat{R} = \widehat{R}_1$ will be preferentially ionized.

This phenomenon is similar to the famous “lock-and-key” mechanism which enables metabolic reactions by allowing biological molecules to recognize right partners through fitting their 3D shapes. In our case, two dimensions are fixed by the light’s polarization plane, while the third direction is due to time and is enabled by our ability to excite and resolve electronic coherences. Since the match-making between the electron current and sub-cycle evolution of the laser field leads to preferential ionization of specific molecular orientations, the cations produced through dissociative ionization will be selectively oriented as a function of the time delay (Fig 4 (e)). It leads to an anisotropic ejection of fragments in case of subsequent dissociative ionization, which appears in our molecular dynamics calculations. Since the driving electron current is chiral, this anisotropy shows up along the light propagation axis, and is assessed in terms of the forward/backward fragment asymmetry (FBFA) displayed in (f). Such dissociative ionization is a clear evidence of enantio-selective charge-directed reactivity.

Fig. 4: (d) Snapshots of the electronic current induced by the pump pulse, on a Rydberg sphere of 10 a.u. radius surrounding the molecule for two distinct orientations $\hat{\mathbf{R}}_i$. The current is defined in the molecular frame (equation (4)) and ionization by the probe pulse is enhanced for the molecular orientation where (i) the current co-rotates with the circularly polarized probe field (red arrow) and (ii) the rotation axis of the current aligns with the light propagation vector. This happens here only for $\hat{\mathbf{R}}_1$. (e) Active orientation of the produced cations along the light propagation axis $\hat{\mathbf{z}}$ as a function of time for $\epsilon = 250$ meV. This orientation is defined as the mean value of $\cos \theta_{\text{ion}}$, where θ_{ion} is the angle between the internal C-C bond of the ML cation and $\hat{\mathbf{z}}$, as shown in the inset (see equation (6)). (f) The resulting forward/backward fragment asymmetry along $\hat{\mathbf{z}}$ in the reactive fragmentation of ML cations, following probe-induced ionization of the transient 3d-4p electron wavepacket leading to photoelectrons with energy $\epsilon = 250$ meV (see equation (7)). The insets illustrate the preferential directions of emission of CO_2CH_3 and CH_3CHOH^+ fragments.

This is the new Fig. 4 (d-f) in the revised manuscript.

Together with our experimental results on MP-PECD driven by electronic coherences, this demonstration makes us confident that this work requires high visibility and dissemination that will have an important impact for future research exploiting chiral properties of matter at the molecular scale.

Finally, we thank again the referee whose comments have allowed us to improve our manuscript and to enlarge its scope. We hope that our revised manuscript will fulfill with the referee's requirements for publication in Nature.

Referee #2 (Remarks to the Author):

In the manuscript “Ultrafast chiroptical switching in UV-excited molecules” by Wanie et al., the authors report a combined experimental and theoretical study of electron dynamics in neutral methyl-lactate, a chiral molecule. The authors have used a traditional pump/probe approach to (1) create a coherent electron wave-packet in the bound states (Rydberg states) using a few-femtosecond UV pulse and to (2) probe its dynamics with a time-delayed circularly-polarized IR field, through a photoionization process. The electron dynamics is revealed by the measurement of the angle-resolved photoelectron energy distributions as the time delay between the pump and probe pulses is varied. From the forward-backward asymmetry along the light propagation observed in the photoelectron angular distributions, generally known as PhotoElectron Circular Dichroism (PECD), the authors reports an oscillatory change of direction of the photoelectron emission as the time delay between the pump and probe is varied on a femtosecond timescale. Theoretical calculations performed within the frozen-nuclei approximation link such an oscillatory pattern to a quantum interference process between Rydberg states coherently populated by the pump pulse. In addition, calculations show that the launched electron wave-packet preserves its coherence over tens of femtoseconds.

Overall, this is a very interesting paper which clearly demonstrates that tracking electron dynamics in large molecular systems is slowly coming of age. I have some comments about the authors’ claims, though, but I do think publication of this paper would provide interesting results to the ultra-fast community.

We thank the referee for reviewing our manuscript and are pleased to read that she/he supports the publication of the work in Nature. We are also genuinely thankful for his/her thorough comprehension of our work. We address below the modifications made to the text following his/her recommendations.

Comments:

1-In the introduction, the authors claims that: “Our findings establish a general method to investigate electron dynamics in a variety of chiral systems with high sensitivity and pave the way to a new scheme for enantio-sensitive charge-directed reactivity in neutral chiral molecules.” I don’t feel that this statement reflects correctly the results of this study. It is true that the present study offers very interesting results about electron dynamics in neutral chiral molecules. But, I would argue that the method used in the experiment is quite traditional. The use of femtosecond pump/probe pulses to reveal electron and/or nuclear dynamics in molecular systems has already been reported many times in the literature on a vast number of different systems. I would then suggest to reword this statement to better reflects the originality of the study.

We agree with the referee that the first part of the sentence was misleading as we do not wish to claim that the experimental technique is novel but rather that the unique characteristics of our UV pump pulses are essential for conducting the study reported, as further detailed in the reply to point 3. For this reason, we rephrased the paragraph describing the experiment in the

main text which now provides the exceptional time-resolution we employed compared to all previous pump-probe TR-PECD experiments: “First, a linearly polarized UV pulse promptly launches a coherent electronic wavepacket just below the ionization threshold in the bio-relevant molecule via a two-photon transition. Then, a time-delayed circularly polarized near-infrared (NIR) probe triggers ionization from the transient wavepacket, providing an exceptional instrument response function of 2.90 ± 0.06 fs.”

All previous experiments used conventional UV sources providing ~100 femtoseconds duration, which prevented the observation of ultrafast coherent electron dynamics. Consequently, our current study marks an important entry into the field of few-femtosecond TR-PECD, thanks to the remarkable temporal resolution offered by our UV light source.

2-In the second paragraph of the introduction (line 43-44), the authors claim that “electron dynamics in biochemically-relevant neutral molecules in general remains to be demonstrated”. Shortly after (line 50-51), they also claim that “Seminal pump-probe experiments using attosecond light pulses have revealed electron-driven charge migration in biomolecules”. That seems somewhat contradicting claims. I would recommend to clarify this point.

Electron-driven charge migration in an *ionized* biomolecule was observed for the first time by some of the authors in Ref. 12 [Science 346, 336 (2014)] of the initial manuscript. Both works reported in line 50-51 (Refs. 12 and 17 of the initial manuscript) investigated electron dynamics in the molecular cation, i.e., following the perturbative photoionization of the target by an attosecond extreme ultraviolet pump pulse. In contrast, our work presents the first experimental observation of electron dynamics in a *neutral* bio-relevant chiral molecule, that is following perturbative photoexcitation of the system below the ionization threshold. We have rewritten the second paragraph of the introduction to clarify this point.

3-In the third paragraph of the introduction (line 54-63), the author claims that “Two important technological challenges must be addressed. First, the pump pulse must have well defined characteristics: (i) a photon energy below the ionization threshold, (ii) a broadband energy spectrum to trigger electron motion among multiple electronic states and (iii) a time duration that provides a prompt excitation before any nuclear motion can take place, together with sufficient temporal resolution” Even though all of these characteristics are indeed required for this study, I would not describe this as an important technological challenge, as the generation of a third harmonic of an IR femtosecond pulse is quite trivial.

We have modified the first sentence in line 54 to “Instead, investigating the light-induced electron dynamics of biochemically-relevant chiral molecules in their *neutral* states with high temporal resolution requires new experimental approaches, and important considerations must be taken into account.”.

So far, no experimental work demonstrated the use of few-femtosecond deep-UV pulses (i.e., 266 nm central wavelength with >500 meV bandwidth) providing the required characteristics to study electron dynamics in the neutral states of chiral molecules. Producing such pulses has been an outstanding challenge, since it would unlock investigations of ultrafast (~1 fs) electron

dynamics in complex molecules of biological relevance, as one of its many important applications in ultrafast science. Producing such ultrashort UV radiation using the third harmonic generation (THG) of the IR pulse imposes severe technical restrictions. It cannot be easily achieved by using nonlinear crystals over a large bandwidth because of both phase matching conditions and the intrinsic temporal dispersion induced by the generating medium. Instead, noble gases must be employed at a cost of a lower conversion efficiency. To produce sufficient UV radiation, we perform THG in a highly pressurized gas cell (7.2 bar of neon). Additionally, because the UV radiation must propagate under vacuum to prevent further temporal dispersion (in air the GVD at 266 nm is about $0.10172 \text{ fs}^2/\text{mm}$) the residual gas must be rapidly evacuated so that the propagation is conducted in vacuum. We developed a differential pumping system that allows us to embed the gas cell inside our vacuum pump-probe beamline [see *Opt. Lett.* 44, 1308 (2019)]. Last, the separation of the ultraviolet radiation from the fundamental IR field after generation is another challenge that the ultrafast community is currently facing. We believe that these technical challenges have prevented any experiment exploiting such short UV pulses to be reported until now and that the setup we have developed to enable such studies is a new technology, even though THG is fundamentally a known process.

Although our intention with the revised manuscript is to claim that the novelty of our work resides in the observation of coherent electron-driven dynamics in chiral molecules, it is also important to emphasize that rather than the UV-pump IR-probe approach that we apply to TR-PECD, the originality of our experimental technique in comparison to all previous works resides in the use of ultrabroadband few-femtosecond UV pulses to induce electronic coherences and simultaneously achieve the unprecedented time-resolution required by our study. As the previous version of our manuscript was misleading in regards to the originality of our work compared to previous studies, we made sure to include the time resolution of our experiment both in the abstract and in the introductory paragraph describing our experiment. We have also added to the supplement the transient signal of the parent molecular ion with mass 104 a.u. that was acquired simultaneously to the electrons (new Fig. S1c in the SI). The fit of this signal was used to determine the instrument response function of $2.9 \pm 0.06 \text{ fs}$. The figure is shown below.

This is Fig. S1(c) in the revised submission.

4-Line 112-113: “which captures an inversion of the photoelectron forward-backward asymmetry in about 7 fs.” I would suggest to be more specific with both the description of Fig 1.b and the figure itself as it is not directly obvious to see this oscillation in the asymmetry parameter. Part of the PECD image does not seem to oscillate at all from 11 to 26fs. The magnitude does change but not the direction of emission.

We thank the referee for this remark as it allowed us to enhance our figure in order to ease the reading of the angularly-resolved PECD plots of Fig. 1(b), making it more accessible to the reader. Fig.1b is now presented as Fig. 1c in the revised manuscript (see below). We included white dashed circles to delineates the kinetic energy of 100 meV, enabling direct observation of the inversion of the electron emission at lower kinetic energies. This energy range below 100 meV is the first one that we consider later on in Fig. 2. We also added to the new Fig. 1b a simplistic representation of the inversion effect observed in the measurements, in order to prepare the reader to identify it in Fig. 1c. The entire revised figure is shown below.

Fig. 1: **Light-induced chiral dynamics of methyl-lactate.** (a) A few-femtosecond linearly polarized UV pulse excites an ensemble of randomly oriented chiral molecules, creating an electronic wavepacket of Rydberg states via 2-photon absorption. The dynamics is probed via 1-photon ionization by a time-delayed circularly polarized NIR pulse. The probing step leads to the ejection of photoelectrons along the light propagation axis defined along the z direction and the resulting angular distribution is recorded by a velocity map imaging spectrometer. (b) The red and blue structure shows the temporal evolution of the coherent electron density in the excited neutral molecule: the chiral evolution of the photoexcited Rydberg wavepacket leads to a reversal of the 3D photoelectron angular distribution at two distinct time delays t and $t + \Delta t$, captured by the measurements. (c) For each time delay, an image is recorded for both left and right circular polarization of the probe pulse. The differential image PECD(ϵ, θ, t) defined in the main text is shown for time delays of 5, 11, 17 and 26 fs for photoelectrons with kinetic energies from 25 to 300 meV along the radial coordinate. The white circles identify the photoelectrons below 100 meV which experience an ultrafast reversal of their emission direction.

This is the new Fig. 1 of the revised version.

5-Regarding the theoretical model, It is true that a full quantum calculation is probably not feasible, and that classical molecular dynamics are probably the only tool available. But, I feel that arguments justifying how classical molecular dynamics are appropriate to extract the physics for this particular system are missing in the main text. Authors should probably further comment on the use of this tool.

We added the adequacy of ab initio (on-the-fly) classical molecular dynamics in the Methods section of our revised manuscript. Indeed, we do not implement quantum-mechanical molecular dynamics calculations – this type of calculations is not included in the Newton-X software that we use. However, sudden excitation and ionization processes mainly occur in classically allowed regions of the potential energy surfaces. The nuclei stay in these regions for short pump-probe delays of a few tens of femtoseconds, which is the timescale of interest in our work. Shortcomings related to the classical nature of the calculations could arise at longer times, when nuclear dynamics involves electron potential barriers along the reaction path. As a matter of fact, we infer that such a barrier shows up in the fragmentation of methyl-lactate cations, $C_4H_8O_3^+ \rightarrow CO_2CH_3 + CH_3CHOH^+$, since the fragments stay close to each other for some hundreds of fs along various nuclear trajectories. This is described in Section 2.4.2 of the SI. However, this is not presently a relevant issue since we are not interested here in large-scale nuclear dynamics, beyond the demonstration of the forward/backward fragment asymmetry (FBFA) effect that is newly introduced in our revised manuscript.

Indeed, to further highlight the impact of our findings we added a new section (including Fig. 4d-f) in our revised manuscript to show how the chiral currents generated by the coherent superposition of excited electronic states can act as an enantio-selective filter of molecular orientations in the laboratory frame upon photoionization by a circularly polarized probe. We address the experimental feasibility to demonstrate such effect by using our trajectory simulations.

6-Lastly, the authors should probably provide information regarding the state of the molecule before irradiation. As described in the experimental method section, the molecules are generated via evaporation. Is there any possibility to have some molecules already populated in Rydberg states before irradiation? If yes, how this pre-population would affect the findings? Following the same idea, the more likely the evaporation process would lead to some excited vibrational states of the molecules. Could the author quantify that and comment on how this would affect the electron dynamics?

Our liquid sample was evaporated and transported by diffusion using a temperature gradient with a final temperature of 95°C at the needle interfacing with the skimmer of our VMI spectrometer. We have included the details of this temperature gradient in the Methods section of the main text. Consequently, the upper limit for the internal temperature of this molecular beam is in the condition of thermal equilibrium with the needle, corresponding to ~ 0.032 eV. The lowest vibrational mode of methyl-lactate for the most stable conformer is the C-C bond stretching of the carbon alpha at 637.5 cm^{-1} , i.e., 0.079 eV [Vibrational Spectroscopy 36 (2004)], which is more than twice the internal temperature of our molecular beam. Therefore, we can consider the population of this lowest energy state negligible.

Furthermore, there is no initial population of excited vibrational nor Rydberg states (these later ones laying above 8 eV – see Fig. S6 of the SI).

Finally, we sincerely thank the referee whose comments have allowed us to substantially improve our manuscript.

Reviewer Reports on the First Revision:

Referee #2 (Remarks to the Author):

The authors have successfully addressed all my concerns and significantly enhanced the quality of their manuscript. From a scientific point of view, I do recommend this manuscript for publication in Nature.

Referee #3 (Remarks to the Author):

The authors describe an impressive experiment demonstrating the ability of time-resolved photoelectron circular dichroism to distinguish electron dynamics in a chiral molecule. I have been watching the theoretical and experimental developments in this area for some time now, and have been waiting for something like this work to be reported. It is gratifying to see it. The authors have used a very short UV pump pulse to create a Rydberg wavepacket. Since the molecule is chiral, the ionization propensity depends on the handedness of an ionizing circularly polarized probe pulse. The authors show that this dependence changes with time due to the evolution of the electronic wavepacket. They further show that the observed oscillations are (at least qualitatively) reproduced by computations. This is an exciting paper which will open up studies of ultrafast dynamics in chiral molecules.

Others have explored the possibility of using circularly polarized light for the excitation pulse (e.g. Huck et al, Science 273 1686 1996 and Raucci et al Nature Comm 13 2091 2022) but the main difficulty has been the relatively low (in absolute terms) enantiomeric excess of the excitation. Similar issues seem to occur here, since the oscillations in the signal seem to be rather small even when the sample is enantiomerically pure (or nearly so). Some comment on this would be appropriate. Nevertheless, the current work shows that one can access the chiral-sensitive signal on an ultrafast time scale, and this is a major advance.

I have a few comments which should be addressed prior to publication.

On page 3, the authors say “strong-field multiphoton driven processes that do not exist in nature.” They might rather say “that are rarely relevant with naturally-occurring light sources.”

On page 5, the authors refer to Fig S4 of SI, but I think they mean S5.

On page 5 (and later in the methods section), the authors speak of “large-scale time-dependent density functional theory.” I have no idea what “large-scale” means and this should be removed.

On page 8, the authors discuss the possible mechanisms for decoherence (expected to lead to the damping of the signal observed in Figures 3b,3d). They argue that nuclear motion is not relevant because the excited state surfaces remain parallel (for ground state cation dynamics). But they then implicate internal conversion, which also is promoted by near or true crossing of electronic states

(and thus downweighted by the very same arguments). The discussion about possible origins of the decoherence is fine, but the calculations intended to explore the origins are not really very persuasive. At the same time, I am not surprised that the decoherence is observed and I don't think there is any compelling reason to try to explain the origin – as the authors themselves note, it would be a serious undertaking to do this (going beyond ground state cation dynamics and including electronic state transitions).

In several places (including page 12), the authors speak of “transient chirality.” This makes no sense to me – the molecule is either chiral or it is not. I guess the authors are trying to say that they are performing a chiral-sensitive measurement with time resolution. That is definitely not what “transient chirality” implies.

It is quite annoying that the authors use at least four different flavors of density functional theory throughout the manuscript: B3LYP, LCBLYP, CAMB3LYP, and X-alpha. Really, they should have been using LCBLYP (or another range-separated functional with full strength exact exchange at long range) for everything. I assume that this does not affect the results, but the authors should verify that.

Author Rebuttals to First Revision:

Below is the point-by-point reply to the referees' comments:

Referee #2 (Remarks to the Author):

The authors have successfully addressed all my concerns and significantly enhanced the quality of their manuscript. From a scientific point of view, I do recommend this manuscript for publication in Nature.

We thank the referee for his/her time in evaluating our revised manuscript and for recommending it for publication in Nature.

Referee #3 (Remarks to the Author):

The authors describe an impressive experiment demonstrating the ability of time-resolved photoelectron circular dichroism to distinguish electron dynamics in a chiral molecule. I have been watching the theoretical and experimental developments in this area for some time now, and have been waiting for something like this work to be reported. It is gratifying to see it. The authors have used a very short UV pump pulse to create a Rydberg wavepacket. Since the molecule is chiral, the ionization propensity depends on the handedness of an ionizing circularly polarized probe pulse. The authors show that this dependence changes with time due to the evolution of the electronic wavepacket. They further show that the observed oscillations are (at least qualitatively) reproduced by computations. This is an exciting paper which will open up studies of ultrafast dynamics in chiral molecules.

Others have explored the possibility of using circularly polarized light for the excitation pulse (e.g. Huck et al, Science 273 1686 1996 and Raucci et al Nature Comm 13 2091 2022) but the main difficulty has been the relatively low (in absolute terms) enantiomeric excess of the excitation. Similar issues seem to occur here, since the oscillations in the signal seem to be rather small even when the sample is enantiomerically pure (or nearly so). Some comment on this would be appropriate. Nevertheless, the current work shows that one can access the chiral-sensitive signal on an ultrafast time scale, and this is a major advance.

We thank the referee for the very constructive comments, which helped in further improving the quality of presentation of our manuscript. The modifications to the text are described in point-by-point reply to his/her comments reported below.

Regarding the enantiomeric excess of excitation, we point-out that our pump pulse is achiral and thus will not preferentially excite one of the two enantiomers. The PECD signal solely relies on electric dipole interactions and results in a remarkable amplitude of the observed oscillations, on the order of 10 % in contrast to common 10^{-2} - 10^{-3} g-factor achieved in Huck et al, Science 273 1686 1996 and Raucci et al Nature Comm 13 2091 2022. The observed contrast is much stronger than any signal that could potentially be detected with conventional circular dichroism, which instead relies on the product of strong electric dipole and weak magnetic dipole interactions. While optimal contrast is expected for a pure enantiomeric sample, the enantiomers comparison reported in Extended Data Fig. 4 shows that our observations are not significantly affected for a difference in enantiomeric excess of 1%.

I have a few comments which should be addressed prior to publication.

On page 3, the authors say “strong-field multiphoton driven processes that do not exist in nature.” They might rather say “that are rarely relevant with naturally-occurring light sources.”

We thank the referee for this suggestion. We have modified the text as follows: ‘that rarely occur with natural light sources.’

On page 5, the authors refer to Fig S4 of SI, but I think they mean S5.

We thank the referee for identifying this typo. In the revised manuscript we now properly refer to Extended Data Fig. 4.

On page 5 (and later in the methods section), the authors speak of “large-scale time dependent density functional theory.” I have no idea what “large-scale” means and this should be removed.

As suggested, we have removed the term ‘large-scale’ for clarity.

On page 8, the authors discuss the possible mechanisms for decoherence (expected to lead to the damping of the signal observed in Figures 3b,3d). They argue that nuclear motion is not relevant because the excited state surfaces remain parallel (for ground state cation dynamics). But they then implicate internal conversion, which also is promoted by near or true crossing of electronic states (and thus downweighted by the very same arguments). The discussion about possible origins of the decoherence is fine, but the calculations intended to explore the origins are not really very persuasive. At the same time, I am not surprised that the decoherence is observed and I don’t think there is any compelling reason to try to explain the origin – as the authors themselves note, it would be a serious undertaking to do this (going beyond ground state cation dynamics and including electronic state transitions).

We agree with the referee’s comment. In the revised manuscript, we have now substantially shortened the discussion dedicated to the decoherence processes. In the main text we now only state that our calculations *suggest* that the *most probable* source of decoherence is the non-adiabatic transitions and we refer to the SI (in which we still include our *tentative* molecular dynamics calculations).

In several places (including page 12), the authors speak of “transient chirality.” This makes no sense to me – the molecule is either chiral or it is not. I guess the authors are trying to say that they are performing a chiral-sensitive measurement with time resolution. That is definitely not what “transient chirality” implies.

Indeed, the molecule is always chiral from its stereochemical description and thus we agree with the referee that the term ‘transient chirality’ can be misleading. Our observations capture a dynamical electronic chiral response that changes over time while the position of the nuclei remains largely unchanged, therefore not affecting the stereochemistry of the

molecule. For sake of clarity, and as suggested by the referee, we have now removed the term 'transient chirality' from the text.

It is quite annoying that the authors use at least four different flavors of density functional theory throughout the manuscript: B3LYP, LCBLYP, CAMB3LYP, and X-alpha. Really, they should have been using LCBLYP (or another range-separated functional with full strength exact exchange at long range) for everything. I assume that this does not affect the results, but the authors should verify that.

We acknowledge the referee's concerns, but using the same hybrid functional throughout our work is not feasible since hybrid functionals such as B3LYP, CAMB3LYP and LCBLYP cannot be employed in the computation of ionization dipoles, which instead necessitates an explicit form of exchange potentials. However, the referee is right when he/she states that LCBLYP could have been employed in all the three remaining stages of our theoretical description (i.e. in the geometry optimizations of both the neutral and cationic forms of methyl-lactate, besides the computation of two-photon absorption tensors) without affecting the results. We illustrate this in the following figure, which shows that the equilibrium geometries of neutral and cationic methyl-lactate are essentially the same in the B3LYP/LCBLYP and CAMB3LYP/LCBLYP frameworks, respectively.

Figure: Equilibrium geometries of neutral (top row) and cationic (bottom row) methyl-lactate (ML) obtained by B3LYP/LCBLYP and CAMB3LYP/LCBLYP DFT approaches, respectively. For cationic ML, we show its second isomer (referred to as Isomer2 in our manuscript) which is involved in ML fragmentation. The lengths of the main bonds are given in Å.

As assumed by the referee, this agreement indicates that our calculations would not be significantly affected by the choice of the functionals employed in the geometry optimization steps.

We sincerely hope that by having addressed all the requests/comments of referee #3, the revised version of the manuscript is now suitable for publication in Nature.